# RL Fine-Tuning Heals OOD Forgetting in SFT

## Abstract

The two-stage fine-tuning paradigm of Supervised Fine-Tuning (SFT) followed by Reinforcement Learning (RL) has empirically shown better reasoning performance than one-stage SFT for the post-training of Large Language Models (LLMs). However, the evolution and mechanism behind the synergy of SFT and RL are still underexplored and inconclusive. To figure out this issue, we dissect the Out-Of-Distribution (OOD) and In-Distribution (ID) reasoning performance of LLM at different checkpoints of the fine-tuning (full-parameter, rather than LoRA) process, and conduct fine-grained analysis. We find the well-known claim "SFT memorizes, RL generalizes" is over-simplified, and discover that: (1) OOD performance peaks at the early stage of SFT and then declines (OOD forgetting), the best SFT checkpoint cannot be captured by training/test loss; (2) the subsequent RL stage does not generate fundamentally better OOD capability, instead it plays an **OOD restoration** role, recovering the lost reasoning ability during SFT; (3) The recovery ability has boundaries, *i.e.,* **if SFT trains for too short or too long, RL cannot improve the OOD ability;** (4) To uncover the underlying mechanisms behind the forgetting and restoration process, we employ SVD analysis on parameter matrices, manually edit them, and observe their impacts on model performance. Unlike the common belief that the shift of model capacity mainly results from the changes of singular values, we find that they are actually quite stable throughout fine-tuning. Instead, the OOD behavior strongly correlates with the **rotation of singular vectors**. In a nutshell, SFT performs hard alignment of the crucial parameter directions to the target tasks, leading to **rapid and greedy adjustment, but also quick forgetting**; RL then **conditionally re-aligns singular vectors softly and slowly** towards a more robust configuration, healing the forgetting and learning the downstream tasks simultaneously. Our findings re-identify the roles of SFT and RL in the two-stage fine-tuning, discover the key mechanism and provide new insights to fine-tuning.

## 1 Introduction

Supervised Fine-Tuning (SFT) is the most widely used method for the post-training of Large Language Models (LLMs) (Howard & Ruder, 2018; Radford et al., 2018). Recent work demonstrates that Reinforcement Learning Fine-Tuning (RLFT), especially when applying after SFT (DeepSeek-AI, 2025), can achieve much better performance on complex reasoning tasks, such as symbolic math reasoning (DeepSeek-AI, 2025; xAI, 2025), code generation (Mirzadeh et al., 2024; Jiang et al., 2024; Anthropic, 2025), embodied tasks (Lin et al., 2025; Li et al., 2025; Zhao et al., 2021), video prediction (Shi et al., 2025), *etc.* Such two-stage fine-tuning paradigm has rapidly become popular because of its advantages over the one-stage SFT (Hugging Face, 2025; Wang et al., 2025).

Numerous studies have explored how RL helps SFT in post-training: a growing body of work argues that SFT tends to memorize or overfit the training distribution, whereas RL yields better out-of-distribution (OOD) generalization (Kirk et al., 2023; Chu et al., 2025); others emphasize that KL-regularized RL counteracts SFT's drift from the base model (Fu et al., 2025), and that rule-based or structure-aware RL can significantly strengthen reasoning (Xie et al., 2025). The authors in (Xie et al., 2025) note that SFT pulls the policy of a model away from its base initialization, and specific RL recipes can boost reasoning. These empirical findings help to partially explore the high-level picture of two-stage fine-tuning, however, the understanding on the synergy of SFT and RL is still inconclusive. In addition, the evolution of OOD performance during the two-stage fine-tuning also lacks a deeper investigation.

To fill the gaps in the above issues, we perform full-parameter SFT and RLFT and study the Out-Of-Distribution (OOD) and In-Distribution (ID) reasoning behaviors of two popular open-sourced models: LLaMA-3.2-11B-Vision (Grattafiori et al., 2024) and Qwen-2.5-7B (Team, 2024). Specifically, we

track their ID and OOD performance at different checkpoints on various reasoning tasks, including the *GeneralPoints, Navigation and Rank-Determinant Computation* tasks. These controlled environments allow us to monitor and disentangle the evolution of model performance and investigate the roles of SFT and RL in the whole process.

During fine-tuning, we observed that: (1) OOD reasoning performance will **peak rapidly in very early stage of SFT** and then degrades slowly as SFT continues. Such **OOD forgetting** is hard to capture by the traditional overfitting detection methods, as the learning curves for ID training/test loss will continue to decline. (2) **RL is not black magic for reasoning**. It can recover the OOD forgetting in SFT but barely surpass its peak performance. The recovery is only effective within a certain range of SFT checkpoints and we identify the shape of advantage distribution as the main cause of it.

To uncover the underlying factors that have high impacts on the fine-tuned models, we analyze the Singular-Value Decomposition (SVD) of parameter matrices and conduct ablation studies on their influence to model performance. Unlike some recent studies (Bartlett et al., 2017; Yoshida & Miyato, 2017; Li et al., 2024b), in our experiments, we notice that the singular values remain essentially constant throughout both SFT and RL stages. Instead, OOD forgetting and recovery highly correlate with the rotations of the singular vectors. In addition, we provide fine-grained layer-wise and top-$k$ analysis on the singular values/vectors and develop insights for SFT to mitigate OOD forgetting.

## 2 PRELIMINARIES

### 2.1 BASIC CONCEPTS AND NOTATIONS

**Self-Attention in Transformer.** Transformers use self-attention to capture global dependencies between each pair of nodes. The attention mechanism is defined as:

$$H = \text{softmax}\left(\frac{QK^\top}{\sqrt{d_k}}\right)V, \text{ where } Q = XW_Q, K = XW_K, V = XW_V \tag{1}$$

where $X$ is input node feature matrix, and $W_Q, W_K, W_V$ are learnable parameter matrices for query, key, and value matrices. An MLP layer is then applied to each row of $H$

$$\text{MLP}(H) = \sigma(HW_{\text{MLP}} + b_{\text{MLP}})$$

**Supervised Fine-Tuning (SFT).** SFT adapts a pre-trained model to a specific task using a labeled dataset $\mathcal{D} = \{(x_i, y_i)\}$ (Howard & Ruder, 2018; Radford et al., 2018). The standard objective is to minimize the negative log-likelihood (NLL) of the target outputs given the inputs:

$$\mathcal{L}_{\text{SFT}}(\theta) = - \sum_{(x_i,y_i)\in\mathcal{D}} \log p_\theta(y_i \mid x_i) \tag{2}$$

**Reinforcement Learning (RL) Fine-Tuning** In contrast to SFT, RL essentially fine-tunes the model by optimizing the policy $\pi_\theta$ based on a reward signal $R(\cdot)$. The general objective is to maximize the expected reward of the actions made by the model

$$\max_\theta \ \mathbb{E}_{x\sim\pi_\theta}[R(x)]$$

The reward function $R(x)$ evaluates the quality of an action $x$ based on desired attributes, like correctness (Ouyang et al., 2022), clarity (Wang et al., 2023), or adherence to rules (Bai et al., 2022). In this paper, we employ Proximal Policy Optimization (PPO) (Schulman et al., 2017), a popular RL algorithm that stabilizes training by optimizing a clipped surrogate objective. The PPO objective is:

$$\mathcal{L}_{\text{PPO}}(\theta) = \mathbb{E}_t \left[\min\left(r_t(\theta)A_t, \text{clip}(r_t(\theta), 1 - \epsilon, 1 + \epsilon)A_t\right)\right] \tag{3}$$

where $r_t(\theta) = \frac{\pi_\theta(a_t|s_t)}{\pi_{\theta_{\text{old}}}(a_t|s_t)}$ is the probability ratio for state $s_t$ and action $a_t$ at step $t$, $A_t$ is the advantage estimate, and $\epsilon$ is a hyperparameter that constrains the policy update step to avoid excessive shift of policy. The advantage $A_t$ measures how much better (or worse) taking action $a_t$ in state $s_t$ is compared to the average action at that state, as estimated by a value function $V_\phi(s_t)$. A common estimator is the *generalized advantage estimation (GAE)* (Schulman et al., 2016), defined as

$$A_t = \sum_{l=0}^{\infty} (\gamma\lambda)^l \delta_{t+l} \quad \text{with} \quad \delta_t = r_t + \gamma V_\phi(s_{t+1}) - V_\phi(s_t),$$

where $\gamma \in [0, 1]$ is the discount factor, $\lambda \in [0, 1]$ controls the bias-variance trade-off, and $r_t$ is the reward at step $t$. Intuitively, $A_t$ is positive when an action yields higher return than expected and negative otherwise, guiding PPO to reinforce beneficial actions while discouraging harmful ones.

**Singular Value Decomposition (SVD)** For a parameter matrix $M \in \mathbb{R}^{m \times n}$, its SVD is given by:
$$M = U\Sigma V^\top$$
where $U \in \mathbb{R}^{m \times m}$ and $V \in \mathbb{R}^{n \times n}$ are orthogonal matrices whose columns are the left and right singular vectors, respectively. $\Sigma \in \mathbb{R}^{m \times n}$ is a rectangular diagonal matrix containing the non-negative singular values, $\sigma_1 \geq \sigma_2 \geq \cdots \geq 0$.

In the context of a neural network, the singular values $\{\sigma_i\}$ are often interpreted as the importance of different representational modes (Bartlett et al., 2017; Schulman et al., 2016; Raghu et al., 2017), while the singular vectors (the columns of $U$ and $V$) define the directions of these modes. SVD on parameter matrices can help us to understand the internal mechanisms of SFT and RL fine-tuning, and investigate their correlation with the ID and OOD reasoning performance of models.

## 3 EVALUATION AND ANALYSIS

In this section, we investigate the evolution of OOD reasoning ability of LLMs by analyzing the model performance at different checkpoints in SFT and RL stages. More specifically, in Section 3.1, we introduce the experimental settings, including the tasks, models and evaluation methods. In Section 3.2, we present the results and conduct detailed analysis on ID and OOD reasoning performance.

### 3.1 EVALUATION SETTINGS

**Task Description** We use *GeneralPoints, Navigation* (Chu et al., 2025) and *Rank-Determinant Computation* (Sun et al., 2025) [1] to evaluate the arithmetic, spatial and cross-concept math reasoning abilities of models. The detailed task descriptions and prompts are shown in Appendix B.1 and we only use *GeneralPoints* **as an example** in main paper. The *GeneralPoints* environment (Chu et al., 2025) is instantiated on top of the *Points24* environment (Zhai et al., 2024). Each state $s$ contains four poker cards, described in text directly. The goal is to produce an equation that equals a target number (24 by default), with four numbers from the cards used only once. Particularly, the cards $'J, Q, K'$ are all interpreted as the same number 10 in the original setting (for training). For example, provided with four cards $[5, 4, 10, 7]$, we aim to output the equation *(7-5)\*10+4* as the desired output.

**Evaluation of OOD Generalization** To disentangle the evaluation of superficial format learning and real arithmetic reasoning ability, we tweak the rule of *GeneralPoints* as (Chu et al., 2025) and test both ID and OOD reasoning performance of models. Specifically, instead of interpreting $'J, Q, K'$ all as the same number 10, the new rule interprets them as $11, 12$, and $13$, respectively. If the model can really obtain arithmetic reasoning ability, they should perform well on such OOD settings. We record the model performance at different checkpoints to show how the ID and OOD generalization abilities evolve. See the ID and OOD evaluation setups of other tasks in Appendix B.1.

**Models and Setup** We use two most popular open-sourced base models LLaMA-3.2-11B-Vision (Grattafiori et al., 2024) and Qwen-2.5-7B (Team, 2024) as the base models. Following the commonly used two-stage pipeline for post-training (DeepSeek-AI, 2025), we first warm-up the model with SFT, and then run RL on top of SFT checkpoint. The format of the prompt is the same as (Chu et al., 2025). For *GeneralPoints*, we follow the setup in (Chu et al., 2025) as our standard setting, which is to run 1100 SFT checkpoints in total for LLaMA-3.2-11B-Vision[2], 800 SFT checkpoints for Qwen-2.5-7B, and then 15 RL checkpoints for both of them. We denote the checkpoints when SFT and RL end as $\text{SFT}_{\text{End}}$ and $\text{RL}_{\text{End}}$. Besides the standard setting, to track the impact of RL on the SFT model more carefully, we apply RL at different SFT checkpoints $\{0, 90, 140, 200, 300, 400, \ldots, 1600\}$, and evaluate the ID and OOD reasoning performance before and after RL [3]. See the computational resources and setups in Appendix B.2. We use PPO in the main paper, and see the results of GRPO in Appendix C.2.

---

[1] See Appendix C.4 for experimental results on other five benchmark datasets.

[2] We adjust the number of checkpoints of SFT from 400 to 1100 as we employ 4 H100 GPUs for SFT instead of 8 H800 in the original paper.

[3] We only use LLaMA with GeneralPoints as an example to do the checkpoint-wise analysis, as the checkpoint sweep across all models and tasks is prohibited due to computational constraints. For example, in RL stage, we need to use 4 H100 GPUs to train for 24 hours or 8 H100 GPUs to train for 12 hours for each single setting of [model, checkpoint, task].

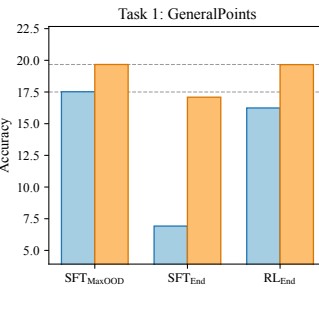 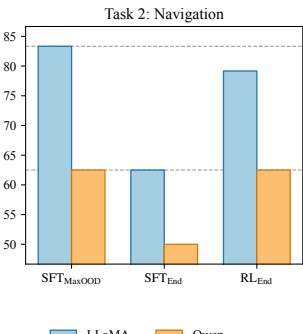 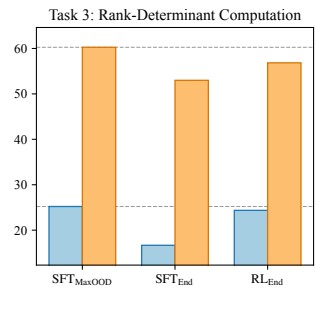

Figure 1: Comparison of OOD performance in three tasks for LLaMA and Qwen at different checkpoints ($SFT_{MaxOOD}$, $SFT_{End}$ and $RL_{End}$).

## 3.2 RESULTS AND ANALYSIS

**What Is Missing in "SFT memorizes, RL generalizes"?** It has recently been found that, in the two-stage fine-tuning pipeline, SFT can stabilize the model output before RL, and RL can enhance the OOD generalization capability of the SFT model (Chu et al., 2025). It highlights the complementary roles of SFT and RL, and the claim "SFT memorizes, RL generalizes" has become popular in AI community. As shown in Figure 1, we managed to reproduce the results in (Chu et al., 2025), where the RL fine-tuned models at $RL_{End}$ significantly outperform models at the checkpoint $SFT_{End}$ on three different tasks. However, when tracking the evolution of OOD performance in the whole SFT process, we can always find a checkpoint (*e.g.,* 140 for LLaMA and 120 for Qwen in GeneralPoints) where the SFT models outperform the RLFT models. This indicates that the conclusion that RL can enhance the OOD reasoning capacity of SFT model is over-simplified and the best overall OOD performance has already been achieved at certain SFT checkpoint. We denote this checkpoint as $SFT_{MaxOOD}$. However, $SFT_{MaxOOD}$ is hard to capture only based on ID training/test losses as shown in Figure 2a. People still tend to manually set up a terminal checkpoint $SFT_{End}$ and then do RL.

LLM keeps losing OOD capability from $SFT_{MaxOOD}$ to $SFT_{End}$. The claim "SFT memorizes, RL generalizes" made in (Chu et al., 2025) is only based on the observations that, starting from $SFT_{End}$, the continued RLFT model is better than SFT model. However, $SFT_{End}$ already suffers from severe OOD forgetting. Therefore, its evidence at a single fixed checkpoint is insufficient to provide a comprehensive and strict comparison between SFT and RL. Their claim reflects only one aspect of a broader picture, where RL recovers the degradation in $SFT_{End}$, but barely surpasses the best of SFT. To depict the whole story and verify our new claim, we track the OOD performance at various SFT checkpoints and apply RLFT (As stated before, we only use LLaMA on GeneralPoints as example due to the prohibitive computational cost of this experiment [4]). Our observations of the whole fine-tuning process are as follows.

**SFT forgets.** The training loss and ID test loss during SFT are shown in Figure 2a, the format loss is shown in Figure 2b, and the OOD and ID test accuracy curve (take LLaMA as an example) is shown in Figure 2c and 2d. As shown in Figure 2b, the format loss converges at checkpoint 50 and stays almost unchanged afterwards, which means the model completes format alignment at $SFT_{50}$. During 50 to 140 checkpoints, the performance gain in OOD reasoning is mainly from the improved arithmetic reasoning ability. As shown in Figure 2c, the OOD test accuracy declines after $SFT_{MaxOOD}$, although the training loss and ID test loss continue to decrease. This performance divergence indicates that the model starts to focus too much on adapting to the rules of the target game, instead of really learning the arithmetic reasoning ability. Such over-specialization causes the model to forget the acquired OOD reasoning ability. Note that we **do not have an overfitting problem** here, because ID test loss keeps decreasing and ID test accuracy continues to increase. However, in this situation, we still keep losing the OOD reasoning ability, and we call such a phenomenon **OOD forgetting**.

**RL recovers.** As shown in Figure 2c, there exits an interval of SFT checkpoints where RL (orange) curve is higher than SFT (green) curve, which means RL can heals the OOD ability of the model

---

[4]See Appendix C.4 for more evidence about the existence of $SFT_{MaxOOD}$ on five more benchmark datasets.

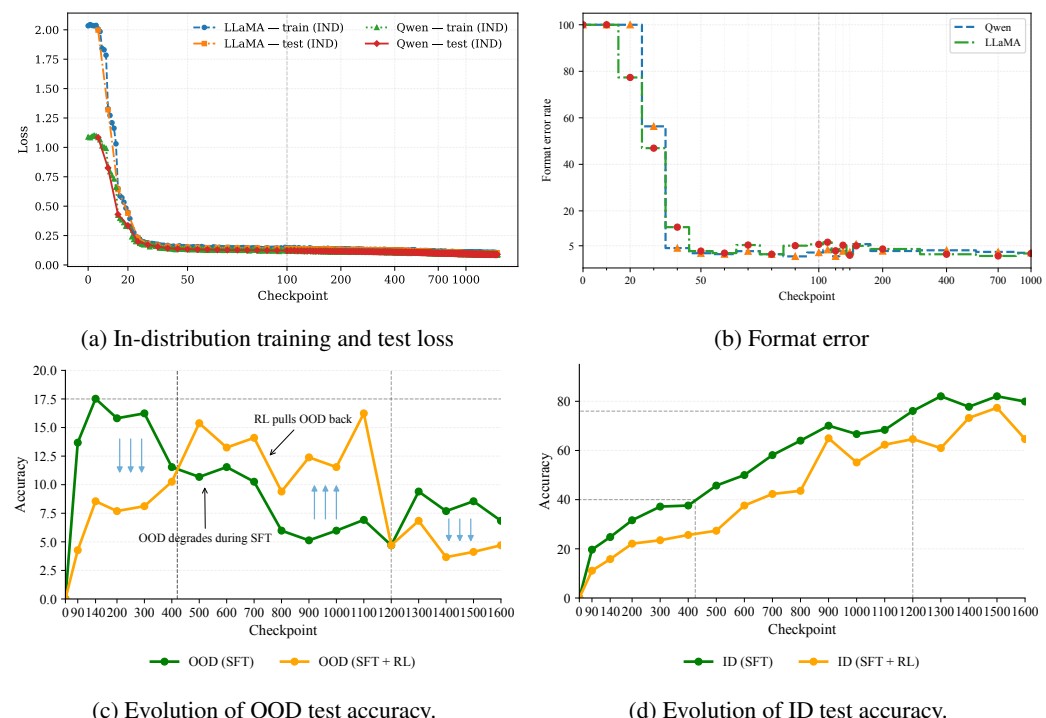

(a) In-distribution training and test loss

(b) Format error

(c) Evolution of OOD test accuracy.

(d) Evolution of ID test accuracy.

Figure 2: (2a) Training and test loss, and (2b) format error curves during SFT. Evolution of (2c) OOD and (2d) ID test accuracy of SFT and RL at different checkpoints (take LLaMA as the main example).

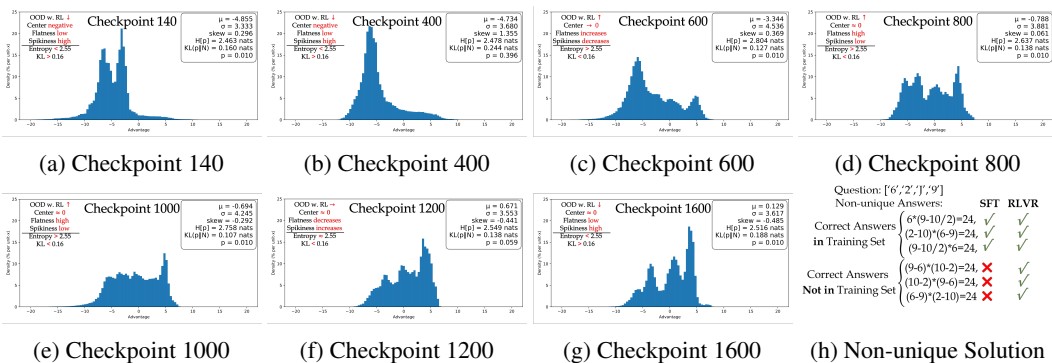

(a) Checkpoint 140    (b) Checkpoint 400    (c) Checkpoint 600    (d) Checkpoint 800

(e) Checkpoint 1000    (f) Checkpoint 1200    (g) Checkpoint 1600    (h) Non-unique Solution

Figure 3: Advantage estimation distribution (a-g) and demonstration of questions with non-unique solutions (h).

that is lost in SFT, with a bit sacrifice of the specialization on ID data [5] (see Figure 2d). Note that, in our experiments, RL cannot help SFT model surpass its peak OOD performance at $SFT_{MaxOOD}$ and fails to generate fundamentally new solutions in most cases, which means that RL barely helps the fine-tuned model escape the constraint of its base model.

Interestingly, there exists **a clear boundary for the recovery effect of RL**, *i.e.,* RL can only restore the lost OOD capability in SFT within checkpoint $[420, 1200]$. The reason is that PPO needs a balanced ratio of positive vs. negative reward signals to be trained stably and effectively. Highly skewed reward distribution can lead to high variance in advantage estimates, poor exploration, and

---

[5] The decrease of ID performance is not due to inadequate training issue in our RL setting, as we follow the same setup as Chu et al. (2025), *i.e.,* we use 4 H100 GPUs to train 24 hours or 8 H100 GPUs to train 12 hours depending on the availability of resources. Continued RLFT will lead to further decline for both ID and OOD performance. Such model deterioration of long-term RL training was also found by other researchers or practitioners. Therefore, we believe that there is no inadequate training problem.

unstable policy updates. Empirically, the proper ratio of positive reward in our experiment can be roughly estimated by the ID accuracy of SFT model as shown in Figure 2d, *i.e.,* around $[40\%, 80\%]$.

**Robustness**    We have verified the robustness of our results with five random seeds at three representative SFT checkpoints (400, 900, and 1600) and do RLFT. The results align with our claim and the variances among seeds are around 1.5, which means that our claim is statistically robust. We also adopted different seeds to evaluate the model and the variance is negligible.

**Advantage Distribution**    To further understand how does the distribution of received rewards impact the effectiveness of RLFT, we go deeper into the advantage estimation distribution, which is important to study the policy update in PPO as shown in equation 3. For better analysis, we calculate some basic statistics of the advantage distributions, *e.g.,* center $\mu$, standard deviation $\sigma$, and skewness (Zwillinger & Kokoska, 1999). Besides, to investigate the distributions more deeply, we use: (1) entropy to measure its flatness and uniformity, and flat distributions without pronounced modes have higher entropy than sharply peaked (concentrated) distributions (Petty, 2018) [6]; (2) KL divergence *w.r.t.* the matched normal distribution [7] to measure its non-Gaussianity (Hyvärinen, 1997; Hyvärinen & Oja, 2000) (or negentropy (Hyvärinen, 2013)), and a large value indicates that it has more structures than a normal distribution [8], *e.g.,* sharper peaks, multiple modes, heavier tails, asymmetry, *etc.*; (3) the p-value of Silverman's test [9] to study its multi-modality (Silverman, 1981).

The results are demonstrated in Figure 3(a-g) (See full results in Appendix C.12). Through the comparison of the statistics, we observe that, for the checkpoints within the effective boundaries of RL **(1)** the centers do not significantly deviate from 0; **(2)** empirically, the entropy is larger than 2.55 and KL divergence against the matched normal distribution is smaller than 0.16, and these two empirical thresholds indicate that the efficacy of RL fine-tuning highly correlates with flat, less spiky and structured advantage signals; **(3)** moderate skewness and multi-modality are acceptable.

Note that our observations of the boundary also echo some empirical observations in recent studies (Liu et al., 2025; Wang et al., 2025) that we need the base model to be strong enough (*e.g.,* more than 420 SFT checkpoints) for RL to be effective; on the other hand, too much SFT (*e.g.,* over 1200 checkpoints) will lead to policy entropy collapse and hurt exploration (Lanchantin et al., 2025).

**Verifiable Reward Shines in Problems With Non-Unique Solutions**    RL recovers the OOD ability lost in SFT by providing better gradient directions through more accurate evaluations, especially for the questions with non-unique solutions. We demonstrate it in Figure 3h and the example in Appendix B.1. As shown in the "number" and "formula" steps, multiple correct formulas can be derived based on the same set of question numbers. However, the token-level cross-entropy loss in SFT will only give "positive reward" to the correct answers that exist in training data. For other newly explored correct solutions, it will give "negative rewards", which provides incorrect gradient directions and leads to high perplexity on them. This is pronounced on reasoning tasks with multiple answers. Therefore, as long as RL can stably work within the boundary, it heals the OOD forgetting.

## 4   ROTATION MATTERS: A SVD ANALYSIS ON PARAMETER MATRICES

Based on our "SFT forgets, RL recovers", found in Section 3.2, we would like to understand what is the underlying mechanism that causes the different behaviors of SFT vs. RL. Recent work has shown that the spectrum of parameter matrices can offer an interpretable window on how its internal representations evolve and how they relate to downstream performance (Staats et al., 2025; Yunis et al., 2024). With this lens, we can track the changes in parameter space during SFT and RL stages with Singular Value Decomposition (SVD) (Aghajanyan et al., 2020; Yunis et al., 2024) and conduct ablation studies to explore the impacts of singular values/vectors of weight matrices on model

---

[6]This is because for a distribution defined on finite discrete support set, the discrete uniform distribution has the maximum entropy (Hyvärinen, 1997).

[7]For checkpoint $k$, suppose the mean and standard deviation of the advantage distribution is $\mu_k, \sigma_k$, then the the matched normal distribution is $N(\mu_k, \sigma_k^2)$.

[8]This is because, given a fixed second moment or variance constraint, the Gaussian distribution achieves maximum differential entropy (Tse, 2017). Among all continuous approximations of the advantage distribution, the matched Gaussian distribution has the least structure, which can be used as a reference to compare.

[9]P-value less than 0.05 indicates significant multi-modality, and p-value greater than 0.05 but less than 0.10 suggests marginal multi-modality (Freeman & Dale, 2013).

performance. We introduce the experimental setup in Section 4.1, present the results and analysis in Section 4.2 and 4.3, and provide actionable insights in Section 4.4 (initial results in Appendix C.5).

## 4.1 SETUP

Based on some recent findings (Staats et al., 2025; Wu et al., 2023; Yuan et al., 2024), which highlight the significance of self-attention parameter matrices in weight adaptation, our analysis focuses on two sets of parameter matrices:

- $W_Q, W_K, W_V$ **in self-attention matrices** are the core components of the self-attention mechanism (Vaswani et al., 2017). They function by projecting the input embeddings into distinct subspaces to compute attention scores and construct context-aware representations.

- $W_{\text{MLP}}$ **in MLP layer** in both LLM models, every MLP block uses an up-projection to widen the hidden state, a gate-projection to apply the SwiGLU gate (Shazeer, 2020), and a down-projection to shrink it back. We did not include the bias term $b_{\text{MLP}}$ in SVD analysis because this term is found to only have minor impact on model performance.

To investigate how does the SFT- and RL-reshaped parameter matrices impact the model performance, we conduct ablation studies on the singular values/vectors of the above parameter matrices (we use the result on *GeneralPoints* as example). Specifically,

- for singular values, we restore the singular values of the fine-tuned parameter matrices, while keep the corresponding singular vectors unmodified, and see if the model performance (OOD forgetting and recovery) will be reverted accordingly. In other words, we roll back $\Sigma_{\text{SFT}_{\text{End}}} \to \Sigma_{\text{SFT}_{\text{MaxOOD}}}, \Sigma_{\text{RL}_{\text{End}}} \to \Sigma_{\text{SFT}_{\text{End}}}$, and evaluate the models with parameter matrices $U_{\text{SFT}_{\text{End}}} \Sigma_{\text{SFT}_{\text{MaxOOD}}} V_{\text{SFT}_{\text{End}}}^\top$ and $U_{\text{RL}_{\text{End}}} \Sigma_{\text{SFT}_{\text{End}}} V_{\text{RL}_{\text{End}}}^\top$ and check the performance shifts.

- Similar to the restoration of singular vectors, we evaluate the model performance with parameter matrices $U_{\text{SFT}_{\text{MaxOOD}}} \Sigma_{\text{SFT}_{\text{End}}} V_{\text{SFT}_{\text{MaxOOD}}}^\top$ and $U_{\text{SFT}_{\text{End}}} \Sigma_{\text{RL}_{\text{End}}} V_{\text{SFT}_{\text{End}}}^\top$.

For LLaMA $\text{SFT}_{\text{MaxOOD}} = 140, \text{SFT}_{\text{End}} = 1100$ and for Qwen$\text{SFT}_{\text{MaxOOD}} = 120, \text{SFT}_{\text{End}} = 800$.

To identify which layers and which set of singular values/vectors play a more important role in OOD forgetting and recovery, we proceed the restoration process step by step according to different layers, and top-$k$ singular values/vectors. More specifically,

- for layer-wise study, we restore the singular values/vectors for every top-$k$ layer, where $k = 5, 10, 15, 20, \ldots, L$ and $L$ is the total number of layers;

- for singular values and vectors, we restore the top-$k$ singular values/vectors for all layers, where $k = 64, 256, 512, 768, 1024, 1536, 2048, 2560, 3072, 3584, (4096$ for LLaMA$)$;

The results are shown in Section 4.2 and 4.3.

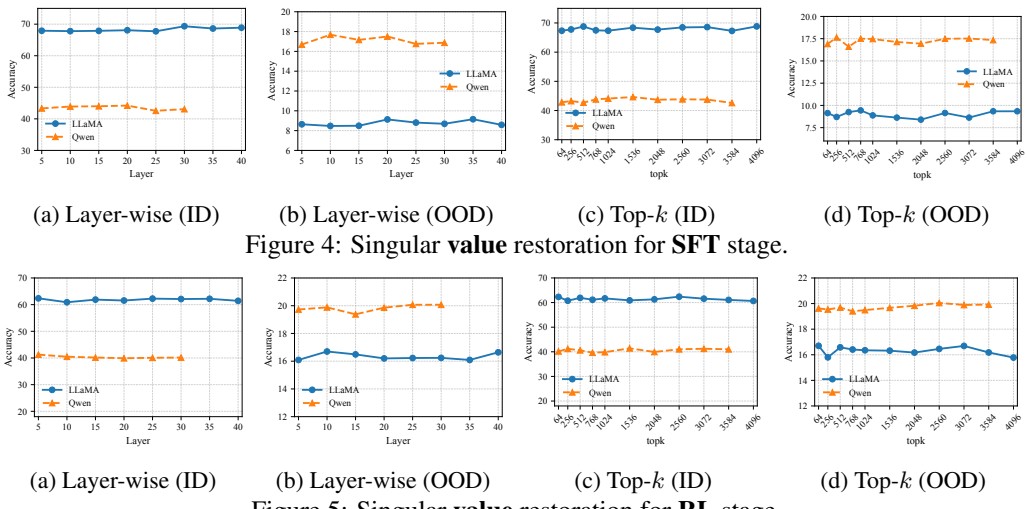

(a) Layer-wise (ID)    (b) Layer-wise (OOD)    (c) Top-$k$ (ID)    (d) Top-$k$ (OOD)

Figure 4: Singular **value** restoration for **SFT** stage.

(a) Layer-wise (ID)    (b) Layer-wise (OOD)    (c) Top-$k$ (ID)    (d) Top-$k$ (OOD)

Figure 5: Singular **value** restoration for **RL** stage.

## 4.2 ABLATION STUDIES ON SINGULAR VALUES

It is found in existing literature that the intrinsic capacity of the model is mainly reflected by the singular values (Bartlett et al., 2017; Yoshida & Miyato, 2017; Li et al., 2024b). However, from our

results of singular value restoration in SFT stage shown in Figure 4, and the results in RL stage shown in Figure 5 [10], we observe that: **the restoration of the singular values of parameter matrices has negligible impact on ID and OOD performance for both SFT and RL fine-tuned models**.

Besides, as the additional evidence shown in Appendix C.8, compared to the original values, the differences of singular values caused by fine-tuning only fluctuate from 0 to 0.005, which act almost as zero-centered noisy signals. This indicates that the fine-tuning process does not significantly amplify or diminish specific singular values. And we do not observe significant shifts concentrated in any particular region, such as the head (largest values) or tail (smallest values), which is found in previous studies (Staats et al., 2025; Thamm et al., 2022; Saada et al., 2025; Cancedda, 2024; Hsu et al., 2022).

### 4.3 ABLATION STUDIES ON SINGULAR VECTOR DIRECTIONS

The results of singular vector restoration in SFT and RL stage are shown in Figure 6 and Figure 7. It is quite clear that **the rotation of the singular vectors plays a more important role than singular values in fine-tuning**, as the ID and OOD performance shift much more significantly. We analyze their fine-grained correlations in SFT stage as follows,

- **Layer-wise Analysis** As shown in Figure 6a and 6b, restoring the singular vectors of first 30 layers of LLaMA and first 15 layers of Qwen causes significant degradation of ID performance. And the restoration of first 10 and last 5 layers leads to the recovery of OOD performance in LLaMA, however, Qwen stays relatively robust. This suggests that, in SFT stage, the task-specific knowledge does not depend too much on the last several layers and OOD capabilities are highly impacted by the the top and bottom blocks of the models.

- **Top-$k$ Analysis** As shown in Figure 6c and 6d, restoring the top 2560 singular vectors of LLaMA and top 2048 singular vectors of Qwen causes significant degradation of ID performance. And the restoration of top 768 singular vectors and last 1024 singular vectors leads to the recovery of OOD performance in LLaMA, however, Qwen stays relatively robust again. This indicates that, in SFT stage, the task-specific knowledge mainly stores in the first several singular vectors and OOD capabilities in the the top and bottom blocks.

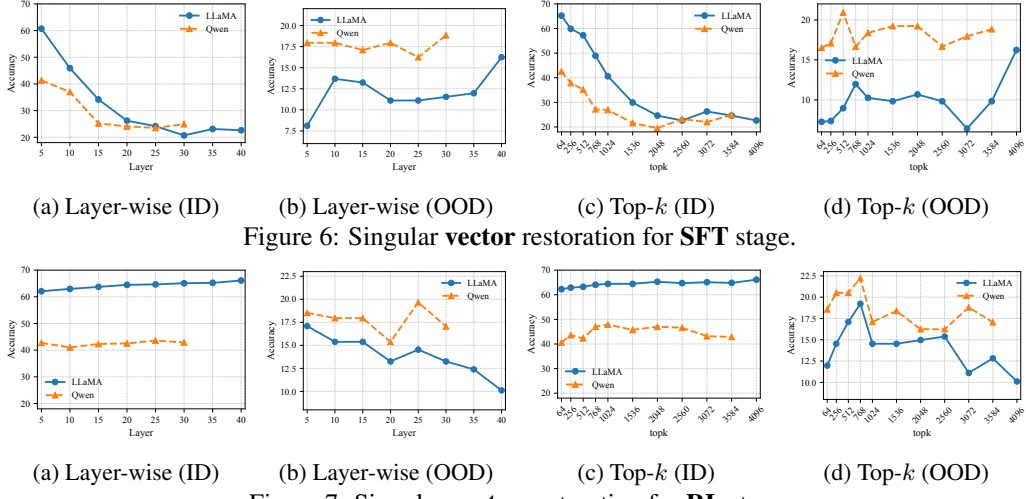

(a) Layer-wise (ID)     (b) Layer-wise (OOD)     (c) Top-$k$ (ID)     (d) Top-$k$ (OOD)

Figure 6: Singular **vector** restoration for **SFT** stage.

(a) Layer-wise (ID)     (b) Layer-wise (OOD)     (c) Top-$k$ (ID)     (d) Top-$k$ (OOD)

Figure 7: Singular **vector** restoration for **RL** stage.

In RL stage, we observe that

- **Layer-wise Analysis** As shown in Figure 7a and 7b, the restoration of singular vectors consistently causes performance degradation of ID and OOD performance for LLaMA, with some perturbations in intermediate $(15 - 25)$ layers for OOD performance. ID and OOD performance of Qwen is relatively robust, and also have some perturbations in intermediate $(15 - 25)$ layers for OOD performance. This indicates that RL uniformly impacts each layers in LLaMA for both task-specific knowledge and OOD ability.

- **Top-$k$ Analysis** As shown in Figure 7c and 7d, the restoration of singular vectors uniformly causes a performance degradation of ID performance for LLaMA, Qwen is relatively robust.

---

[10]See a more detailed study in Appendix C.8

For OOD performance, the top (1024) and bottom ($2560 - 4096$ for LLaMA, $2560 - 3584$ for Qwen) singular vectors are highly relevant.

### 4.4 INSIGHTS

As shown in Figure 6d, the restoration of top singular vectors significantly recover OOD performance in both models. Therefore, we penalize the rotations of singular vectors in the top ranks during SFT without additional RL stage. Compared to vanilla full-parameter SFT, this strategy shows promising results in mitigating OOD forgetting. See Appendix C.5 for details and results.

## 5 RELATED WORK ON RL REASONING

### 5.1 RL IMPROVES REASONING AND OOD GENERALIZATION

Following the introduction of DeepSeek-R1 (DeepSeek-AI, 2025), large-scale RL has emerged as a principal driver of improved reasoning, directly eliciting long chain-of-thought behavior and strong math/coding performance. Notably, the zero-SFT variant (R1-Zero) is trained solely with RL yet already exhibits powerful reasoning. This has motivated work that explicitly disentangles the roles of supervised fine-tuning (SFT) and RL for reasoning and out-of-distribution (OOD) generalization.

Several studies suggest that the two objectives induce different competencies: authors in (Ma et al., 2025) report that RL is more effective on low to medium-difficulty tasks, whereas SFT performs better on harder problems; authors in (Chu et al., 2025) further find that PPO-based RL generalizes better than SFT, which tends to memorize training data rather than acquire transferable reasoning skills. Authors in (Xie et al., 2025) support this claim by demonstrating that rule-based RL enhances LLM reasoning and achieves generalization to challenging benchmarks such as AIME and AMC after training on synthetic logic puzzles (Knights-and-Knaves).

Motivated by the gap between these two paradigms, recent work integrates SFT and RL to improve performance. In particular, UFT (Liu et al., 2025) unifies supervised and reinforcement fine-tuning within a single stage and injects supervision into the RL phase through a hybrid objective. Authors in (Huang et al., 2025) proposes "prefix-RFT", which seeds each rollout with an supervised prefix and trains the continuation with policy-gradient RL.

### 5.2 "COMPARED WITH SFT, DOES RL REALLY HELP?"

On the other hand, there is skepticism about the effectiveness of RL for reasoning. Authors in (Yue et al., 2025) argue that current RLVR mostly improves sampling efficiency rather than expanding a model's reasoning capability boundary, and that at high $k$ (pass@k) base models can outperform their RL-trained counterparts. They conclude that the seemingly "new" reasoning patterns are better attributed to distillation than to RL itself. Authors in (Kim et al., 2025) argue that RLVR does not enhance a model's reasoning ability; rather, it mainly boosts accuracy on easier problems while hurting performance on harder ones. Authors in (Zheng et al., 2025) find that reasoning models augmented by RL significantly underperform their corresponding base models in parody detection tasks, which demonstrate the limitation of RL in general reasoning tasks.

### 5.3 OUR CONTRIBUTIONS

Compared with prior work, we offer a different perspective, which track the evolution and synergy of SFT and RL in reasoning ability. Specifically, we re-investigate the popular claim "SFT memorizes, RL generalizes" and demonstrate its deficiencies. Our results and analysis illustrate that SFT causes the model to lose OOD capability, a phenomenon we name as **OOD forgetting**. RL can only restore the OOD ability lost during the SFT phase, and only within a certain range of checkpoints.

## 6 CONCLUSIONS

In this paper, we study the roles of SFT and RL in the two-stage fine-tuning process and generalize the common belief "SFT memorizes, RL generalizes" to "SFT forgets, RL recovers". More specifically, we found the OOD forgetting issue in SFT stage, the OOD recovery effect in RL stage, and the existence of RL effectiveness boundary. In addition, we observe that RLFT does not endow LLMs with fundamentally new OOD reasoning abilities and rarely surpasses the best OOD checkpoint achieved during SFT. The analysis on advantage distribution reveals that flat, less spiky and structured advantage signals are critical for the effectiveness of RL. We have verified that our claims are robust and generalizable across diverse benchmark tasks and various RL algorithms. SVD analysis further shows that the key factor correlating with OOD forgetting and recovery is not the change in singular values of weight matrices, but the rotation of singular vectors. Based on this, we develop new insight to mitigate OOD forgetting during SFT, which is verified to be effective.

## REPRODUCIBILITY STATEMENT

We have provided the codebase in supplementary material and all the results in this paper are reproducible. Implementation details and experimental setups can be found in Section 3.1, 4.1 and Appendix B

## ETHICS STATEMENT

All of the authors in this paper have read and followed the ethics code.

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

## THE USE OF LARGE LANGUAGE MODELS (LLMS)

Large language models (LLMs) were used in the preparation of this manuscript to improve grammar, clarity, and readability. We also use LLMs to search for related studies.

## A CLARIFICATION: FORGETTING, OVER-SPECIALIZATION, OVER-FITTING, OVER-TRAINING

We would like to clarify the differences between the following concepts to highlight the uniqueness of our study on OOD forgetting and avoid confusion.

- **Catastrophic Forgetting** means that a model loses prior knowledge or skills when trained on new data (Li et al., 2024a; Kotha et al., 2024). More specifically, when we fine-tune an LLM on a new task, it underperforms the original LLM on previously learned tasks/domains.

- **Over-Specialization** refers to format specialization (Wang et al., 2024a), which means that a model becomes narrowly specialized to the format of a task during fine-tuning, even on some inappropriate places. It is a form of forgetting and will lead to failure of OOD generalization. It often happens rapidly at the early stage of forgetting, but may not degrade the deeper knowledge of the LLM.

- **Over-Fitting** happens when a model becomes too much tailored to the training (fine-tuning) data, instead of capturing the general patterns of the corresponding domain. We will observe a decrease in training loss and an increase in validation and test loss. Note that over-fitting is only relevant to the in-distribution generalization settings, but not OOD generalization.

- **Over-Training** happens in model pre-training stage. It means that when pretraining extends too long, even though the base model improves, the post-training performance will drop due to the increased sensitivity to parameter modification (Springer et al., 2025).

## B MORE DETAILS OF EXPERIMENTAL SETTINGS

### B.1 TASK DESCRIPTIONS AND PROMPTS

**GeneralPoints** Prompts and examples for the GeneralPoints game are shown as follows.

```
[Task Description]
You are an expert 24 points card game player. You will receive a set of 4 cards. Note that 'J', 'Q', and 'K'
count as '10', and each card must be used once. Your goal is to output a formula that evaluates to 24 using
numbers from the cards and operators such as '+', '-', '*', '/', and '='.
[Input]
Cards: [1, 3, K, 6]
[Output]
{ "cards": [x, y, z, w], where {face_card_msg},
"number": [a, b, c, d], where a, b, c, and d are the numbers on the cards,
"formula": 'an equation that equals 24',
}
For In-distribution Response:
{
"cards":[1, 3, K, 6],
"number":[1, 3, 10, 6],
"formula":"(10 × 3) − (6 ÷ 1) = 24"
}

For Out-of-Distribution Response:
{
"cards":[1, 3, K, 6],
"number":[1, 3, 13, 6],
"formula":"(6 × (13 − 1)) ÷ 3 = 24"
}
```

**Navigation**   For the Navigation task, which is used in (Chu et al., 2025), we train the model with an absolute direction *e.g.,* turn (source west), then we evaluate the model's OOD performance by relative direction *e.g.,* turn(left). Prompts and examples for Navigation are shown as follows.

---

[Task Description]
You are an expert in navigation. You will receive a sequence of instructions to follow. You are also provided with your observation and action history in text. Your goal is to first analyze the instruction and identify the next sentence to be executed. Then, you need to provide the action to be taken based on the current observation and instruction.
[Instruction]
1. First, turn right to face north.
2. Move forward until you reach next intersection.
3. Turn left to face west.
4. Move forward until you reach next intersection.
5. Turn left to face north.
6. Move forward until you reach next intersection.
7. Turn right to face east.
8. Move forward until you reach next intersection where Levi & Korsinsky, LLP is on your right behind.
9. Turn left to face north.
10. Move forward until you reach next intersection.
11. Turn slightly right to face northeast.
12. Move forward until you reach next intersection.
13. Turn right to face northwest.
14. Move forward until you reach next intersection where Mr Goods Buy & Sell is on your left front.
15. Turn left to face northeast.
16. Move forward until you reach next intersection where Skullfade Barbers is on your left front.
17. Turn right to face northwest.
18. Move forward until you reach destination where The destination Ann Cleaners is on your left.

[Action space]
forward(): indicates moving forward one step
turn direction(x): indicates adjust the ego agent direction towards x direction. x could be any following 8 directions ['north', 'northeast', 'east', 'southeast', 'south', 'southwest', 'west', 'northwest']
stop(): indicates the navigation is finished.
vspace6pt

[Observations and action sequence]
$O_1$: No landmarks nearby;
$A_1$:
For In-distribution Response:
{
"current observation": "No landmarks nearby; "
"current instruction": "First, turn right to face north."
"action": "turn direction(north)"
}

For Out-of-Distribution Response:
{
"current observation": "No landmarks nearby; "
"current instruction": "First, turn right to face north."
"action": "turn direction (right)"
}

---

**Rank-Determinant Computation**   For the matrix computation task, we train LLaMA and Qwen to compute the rank of a matrix with given dimension, *e.g.,*, $4 \times 5$, then we employ the determinant compute as an OOD task to evaluate both models, which is adapted from (Sun et al., 2025). It evaluates not only the math computation, but also the cross-concept math reasoning ability, which is much more complex than the task in (Sun et al., 2025).

For in-distribution training, the prompt is:

```
[Task Description]
You are an expert in linear algebra. You will receive a square matrix. Find the rank of the matrix and output
the integer result.
[Input]
Matrix:[[-1, -2, 9, 3, -5], [0, -3, 9, 9, -6], [-2, -2, 12, 0, -6], [3, -2, -3, 15, -1]]
[Output]
{ "answer": 2,
}
```

For out-of-distribution evaluation, the prompt is:

```
[Task Description]
You are an expert in linear algebra. You will receive a square matrix. Compute its determinant and output the
integer result.
[Input]
Matrix:[[-4, 3], [-3, -2]]
[Output]
{ "answer": 12,
}
```

## B.2  COMPUTATIONAL RESOURCES AND SETUPS

All our RL fine-tuning is implemented on 8xH100 GPUs. SFT utilizes 4xH100 GPUs, the learning
rate is 1e-6, a mini batch size of 64, and cosine is used as the learning rate schedule. We use PPO
with rollout 256 to fine-tune the model after supervised fine-tuning. The checkpoint for different
checkpoints may vary slightly due to the precision or computational resources.

| task | model | MaxOOD | SFT data | RL_begin | #RL ck | RL data | eval data |
|------|-------|--------|----------|----------|--------|---------|-----------|
| GeneralPoint | LLaMA | 140 | 100k | 500–1100 | 15 | 60k | 234 |
| GeneralPoint | Qwen | 120 | 80k | 400–800 | 15 | 60k | 234 |
| Navigation | LLaMA | 45 | 5k | 60 | 15 | 60k | 234 |
| Navigation | Qwen | 95 | 10k | 100 | 15 | 60k | 234 |
| Matrix | LLaMA | 50 | 15k | 140 | 24 | 10k | 234 |
| Matrix | Qwen | 650 | 80k | 800 | 39 | 20k | 234 |

Table 1: Details of implementation configurations.

For more details, please refer to the supplementary material.

## C  MORE EXPERIMENTAL RESULTS

### C.1  ID AND OOD LOSS IN SFT

After 50 checkpoints, we find that the ID and OOD cross-entropy losses go to different directions.
The ID loss approaches 0.15, then keeps stable, and OOD loss increases after the same checkpoints.
However, based on the results in Figure 2c, the OOD accuracy still increases during checkpoint 50
to 140. Such **loss-accuracy discrepancy** exists for both LLaMA and Qwen. After going through
the training and test data as shown in Appendix B.1 during these checkpoints, we found that such
discrepancy is caused by OOD rule forgetting and OOD reasoning enhancement. To be more specific,
after the completion of format alignment at checkpoint 50, the model starts to suffer from over-
specification to the ID rule, failing to turn $'J, Q, K'$ as number $11, 12, 13$, *i.e.,* error in "number" step
in OOD response will increase. The failure of "number" step will be very likely to cause failure in
"formula" step, which will result in large OOD cross-entropy loss. However, during checkpoint 50
to 140, the arithmetic reasoning ability keep improving, *i.e.,* once the model succeed to interpret
$'J, Q, K'$ as number $11, 12, 13$, the model has much higher probability to get a correct "formula".
But compared with the increased loss in both "number" and "formula" steps, the improved accuracy
in "formula" step will only cause a relative smaller decline of loss. So overall, in such mixture of
status, we will observe and increased OOD loss together with increased OOD accuracy. From another
perspective, the loss-accuracy discrepancy tells us that the token-level cross-entropy loss cannot fully
reflect the real reasoning capacity of model.

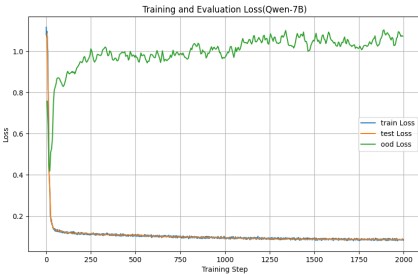 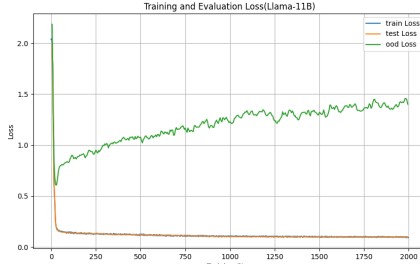

Figure 8: In-distribution training/test loss and OOD loss curves for LLaMA-3.2-11B-Vision and Qwen-2.5-7B during SFT.

## C.2 RESULTS WITH GRPO

We have verified our claims with GRPO (group size = 4) and the OOD and ID results of LLaMA and Qwen on GeneralPoints are shown in Figure 9 and Figure 10. Interestingly, GRPO performs better than PPO in terms of ID performance and worse than PPO for OOD; and overall, both algorithms align with our claim that RL heals OOD forgetting but does not surpass the best of SFT.

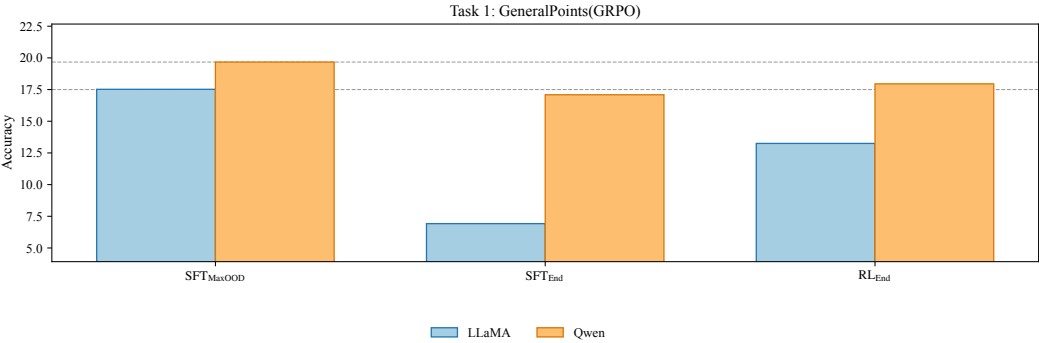

Figure 9: Results of GRPO on GeneralPoints (OOD)

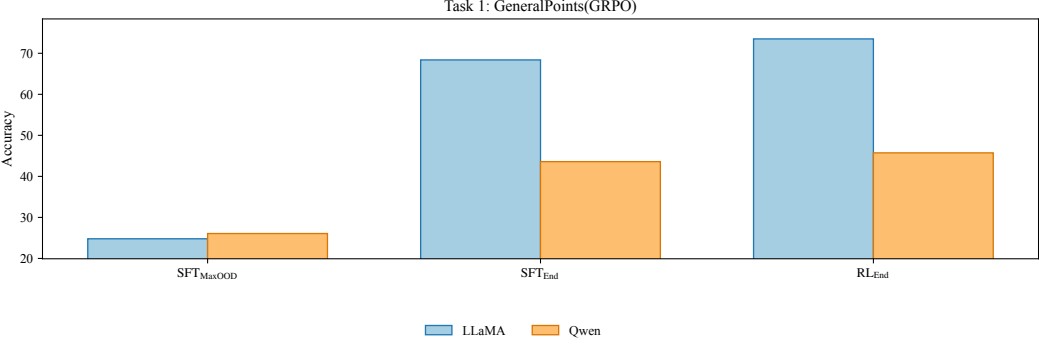

Figure 10: Results of GRPO on GeneralPoints (ID)

## C.3 MORE RESULTS OF IN-DISTRIBUTION GENERALIZATION PERFORMANCE

The in-distribution performance on *GeneralPoints, Navigation* and *Rank-Determinant Computation* are shown in Figure 11.

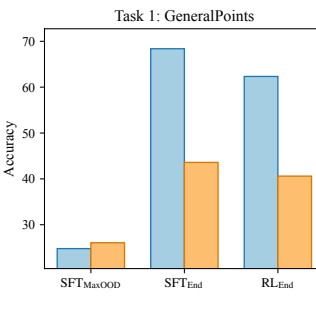 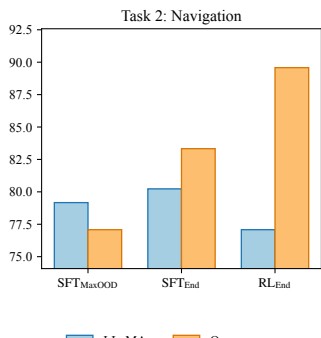 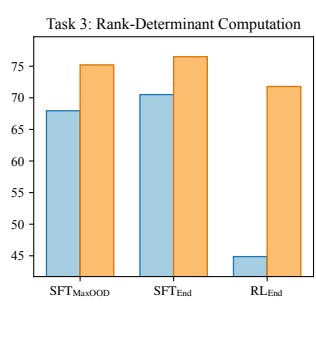

Figure 11: ID performance on three tasks

## C.4 OOD RESULTS ON OTHER BENCHMARK TASKS

| Model/Tasks | | Checkpoints | | |
| --- | --- | --- | --- | --- |
| | | SFT$_{MaxOOD}$ | SFT$_{End}$ | RL$_{End}$ |
| LLaMA | ARC-Challenge | 0.8 | 0.7 | 0.8 |
| | CommonsenseQA | 0.81 | 0.62 | 0.69 |
| | GPQA | 0.44 | 0.32 | 0.44 |
| | IFEval-loose | 0.27 | 0.27 | 0.27 |
| | MMLU-Pro | 0.46 | 0.36 | 0.37 |
| Qwen | ARC-Challenge | 0.92 | 0.88 | 0.88 |
| | CommonsenseQA | 0.79 | 0.73 | 0.75 |
| | GPQA | 0.44 | 0.42 | 0.44 |
| | IFEval-loose | 0.63 | 0.67 | 0.7 |
| | MMLU-Pro | 0.73 | 0.66 | 0.66 |

Table 2: OOD results at checkpoints SFT$_{MaxOOD}$, SFT$_{End}$ and RL$_{End}$ on five more benchmark datasets for LLaMA and Qwen

We have verified our claims on other diverse benchmark datasets: ARC-Challenge Clark et al. (2018), CommonsenseQA Talmor et al. (2019), GPQA Rein et al. (2024), IFEval-loose Zhou et al. (2023), MMLU-Pro Wang et al. (2024b). The results are shown in Table 2 and they align with our observations that the RL heals the OOD forgetting in SFT but barely surpasses the best of SFT. This indicates that our conclusion is robust and generalizable across different tasks.

## C.5 EVOLUTION OF ROTATION-AWARE FINE-TUNING

Inspired by our previous experiments in Section 4, we find that the top singular vectors dominate around 70% of the performance of ID and OOD, and the recovery of singular vectors nearly rolls back the performance for both models to the previous stage. Then we penalize the singular vectors in the top rank (*e.g.,* 128, 256, 512, 1024) to preserve the main directions in high-dimensional space while learning the new task. This strategy reduces catastrophic forgetting compared with the vanilla full-parameter SFT as shown in Figure 12. We plan to expand this method to more generic tasks in future experiments.

## C.6 LOSS OF SINGLE-STAGE RL FINE-TUNING

As summarized in Section 5, there are numerous studies that give completely different conclusions about the effectiveness of RL fine-tuning, especially for single-stage RL. So in this paper, we also verify RL fine-tuning without SFT as cold start.

From Figure 13, we observe that RL can hardly converge without SFT. This is because the base model has poor task-following ability, which would give overwhelmingly low scores for RL, leading to unstable updates and collapse in training. On the other hand, SFT can provide a safe starting point and policy initialization, where the model can at least align the format and generate reasonable candidates for the reward model to evaluate.

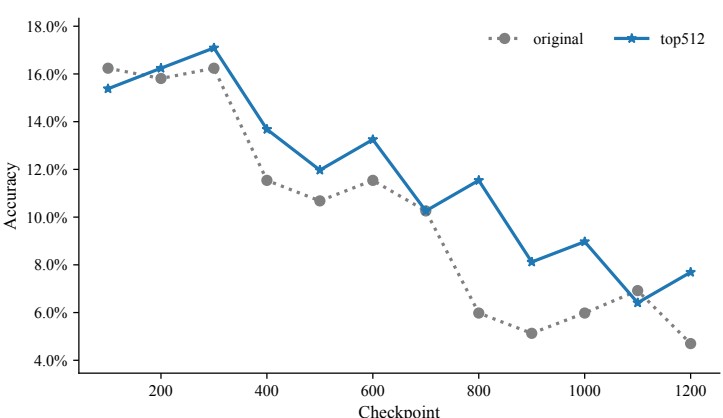

Figure 12: OOD performance after penalty of top 512 rank in singular vectors compared to the original fine-tuning, as shown in the figure, the OOD accuracy is nearly always higher than the original fine-tuning, while maintaining the comparable performance in terms of ID accuracy.

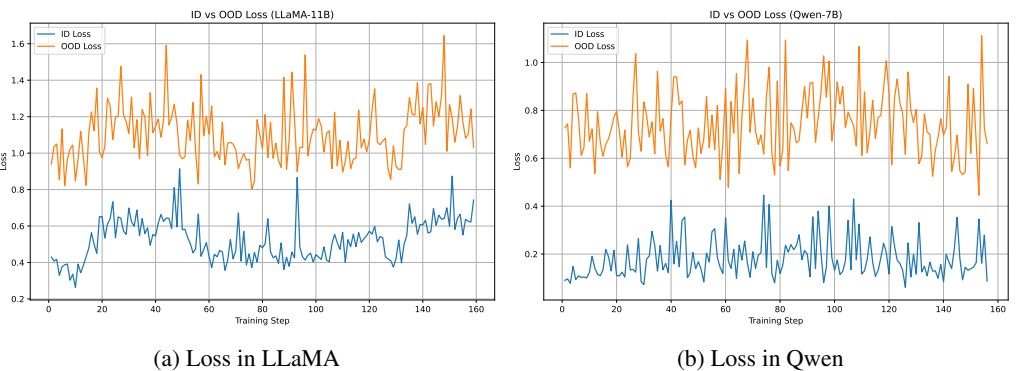

(a) Loss in LLaMA                    (b) Loss in Qwen

Figure 13: Loss of single-stage RL fine-tuning

### C.7 EXAMPLES FOR REWARD HACKING

Inconsistent with previous research (DeepSeek-AI, 2025), as demonstrated below, reward hacking occurs when we fine-tune the models by pure RL from scratch or an early SFT checkpoint.

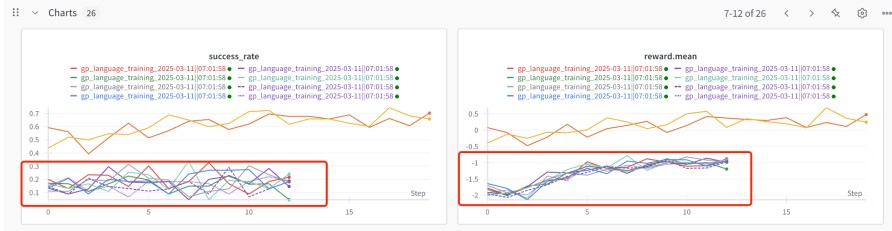

Figure 14: An example of *reward hacking*. The RL-only curve sees an increasing reward signal (right panel) but stagnant or low success rates (left panel).

### C.8 CHANGES OF SINGULAR VALUES

To investigate how does SFT and RL reshape the spectral structure of the parameter matrices, we analyze the singular values of $W_q, W_k, W_v$ and their differences ($\Delta\sigma_i = \sigma_i^{\text{SFT}_{1100}} - \sigma_i^{\text{SFT}_{140}}$ for LLaMA and $\sigma_i^{\text{SFT}_{1100}} - \sigma_i^{\text{SFT}_{140}}$ for Qwen) before/after different training stage. The results are shown in the Figure 15. We found that: **the changes of singular values of the $Q, K, V$ matrices are negligible after both SFT and RL stages across all experiments**. Compared to the original singular values, $\Delta\sigma$ fluctuates from 0 to 0.005, which acts similar as a low-magnitude, zero-centered noisy signals. This indicates that the fine-tuning process does not significantly amplify or diminish specific singular values.

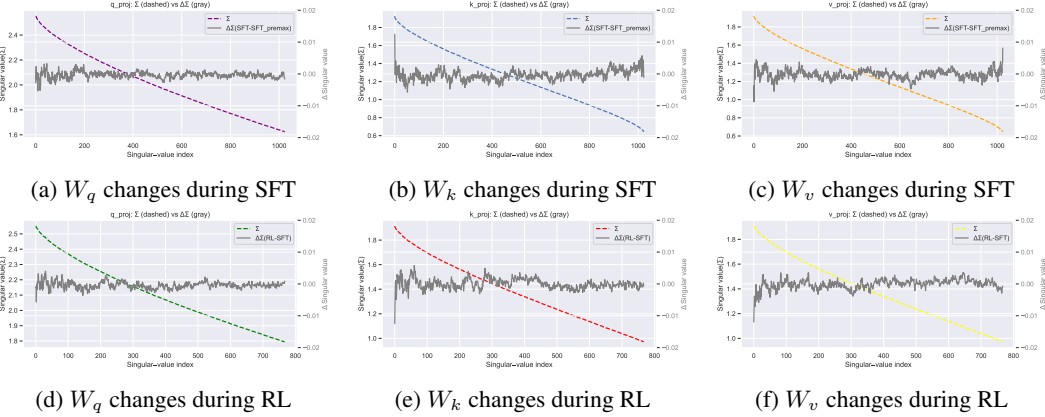

(a) $W_q$ changes during SFT     (b) $W_k$ changes during SFT     (c) $W_v$ changes during SFT

(d) $W_q$ changes during RL     (e) $W_k$ changes during RL     (f) $W_v$ changes during RL

Figure 15: Singular value changes in the q_proj, k_proj, and v_proj matrices of the first self-attention layer (layers[5].self_attn) in LLaMA-3.2-11B-Vision. Panels (a)–(c) illustrate the impact of supervised fine-tuning (SFT) on $W_q$, $W_k$, and $W_v$, respectively, while panels (d)–(f) depict the corresponding changes following reinforcement learning (RL). Each panel shows the difference in singular values before and after the respective post-training stage. For LLaMA, SFT starts from $\text{SFT}_{\text{MaxOOD}}$ (checkpoint 140), RL stage begins from $\text{SFT}_{\text{End}}$ (checkpoint 1100).

### C.9 EXPLORING THE ROTATION OF SINGULAR VECTOR WITH PRINCIPAL ANGLES

There exists two ways to measure the changes of singular vectors during fine-tuning: vector-level metrics and subspace-level metrics.

Principal angles (or canonical angles) quantify how far two subspaces are within the same Euclidean space. To quantify the differences between the subspaces spanned by the singular vectors of base model $W_{\text{Base}}$ and fine-tuned model $W_{\text{FT}}$, we measure the amount of rotations between two subspaces by how much their dominant singular vector directions have *rotated*, which is a commonly used method in machine learning (Huang et al., 2015; Vahidian et al., 2023) and numerical computation

(Björck & Golub, 1973). We provide a brief introduction and we take the left singular vectors for example and the computation includes,

**(i) SVD.** For each matrix, we keep all singular vectors in our experiments,
$$W = U \Sigma V^\top, \qquad U \in \mathbb{R}^{m \times r},\ V \in \mathbb{R}^{n \times r}, \tag{2.4.1}$$
where the columns of $U$ and $V$ are orthonormal and $\Sigma = \mathrm{diag}(\sigma_1, \ldots, \sigma_r)$ with $\sigma_1 \geq \ldots \geq \sigma_r \geq 0$, $r$ is the rank.

**(ii) Computation of Principal Angles Between Subspaces (PABS).** Let $U_{\text{Base}}, U_{\text{FT}} \in \mathbb{R}^{m \times k}$ be the left singular blocks from the previous step. Define $M := U_{\text{Base}}^\top U_{\text{FT}} \in \mathbb{R}^{r \times r}$. Since both of them are orthonormal, the singular values of $M$ lie in $[-1, 1]$ (Björck & Golub, 1973). Suppose the SVD of $M$ is
$$M = U_M \, \mathrm{diag}(s_1, \ldots, s_r) \, V_M^\top,$$
the *principal angles* $\theta_i \in [0, \pi/2]$ between $U_{\text{Base}}^\top$ and $U_{\text{FT}}$ are
$$\theta_i = \arccos(s_i), \quad i = 1, \ldots, r. \tag{2.4.2}$$
The computational complexity is $O(\min\{m, n\}^3)$. An identical procedure on $V_{\text{Base}}, V_{\text{FT}}$ yields angles for the right subspaces. In practice we clamp the numerical values of $s_i$ to $[-1, 1]$ before calling $\arccos$ to avoid floating-point overflow. The Principal angles measure the 'tilt' between corresponding singular vectors of two matrices, *i.e.,* the degree to which two parameter matrices are different from each other in terms of singular vectors under the rank $r$. The angle set $\{\theta_i\}$ serves as a fine-grained measure of subspace rotation: $\theta_i = 0$ means the $i$-th principal direction is preserved, whereas values approaching $\pi/2$ indicate maximal misalignment.

**Advantages of PABS**

- **Numerical Stability:** Consider when two singular values are very close and their corresponding singular vectors are orthogonal. After one step of SFT, the singular values and vectors might only make subtle shifts but the singular values might swap orders. Therefore, the pairwise cosine similarity might demonstrate a very large angle, while the parameter matrices only make subtle changes. Therefore, vector-level metrics are not as robust as subspace-level metrics like PABS.
- Cosine similarity between singular vectors only compares one dimension at a time, without accounting for interdependence between directions. PABS derives angles that reflect the relative orientation of the entire subspace, providing a more informative measure than isolated vector-to-vector comparisons.
- PABS is a true metric for comparing subspaces, ideal for measuring alignment or divergence holistically.

We use principle angle to analyze the pattern of subspace rotation during SFT and RL. To this end, we calculate the principal angle spectrum of the layer-0 $k_{\text{proj}}$ matrix between checkpoint 0 vs. $\text{SFT}_{\text{End}}$, and checkpoint 0 and $\text{RL}_{\text{End}}$, and plot them in Figure 16. For both SFT and RL, the two monotonically increasing curves overlap each other: the smallest angle is around $25 - 30$ degrees and the angles increase smoothly and linearly toward 90 degree in the tail.

These curves imply that both of the two fine-tuning stages adjust the model primarily by rotating its singular vectors, which is already verified in Section 4. However, we cannot find out the differences in their rotation patterns. The exact mechanism of the rotation patterns remains unresolved and understanding the two fine-tuning behaviors in parameter space, especially in high-dimensional space, is an open question that we will investigate in future work.

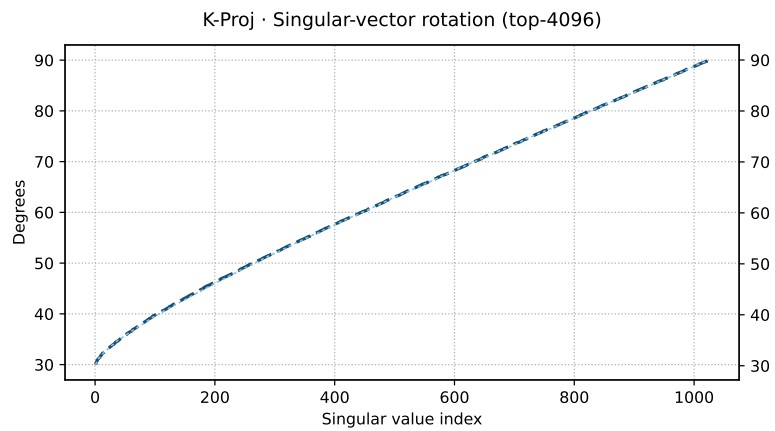

Figure 16: An example of rotation between SFT and RL.

### C.10 PCA VISUALIZATION OF EMBEDDING SHIFTS

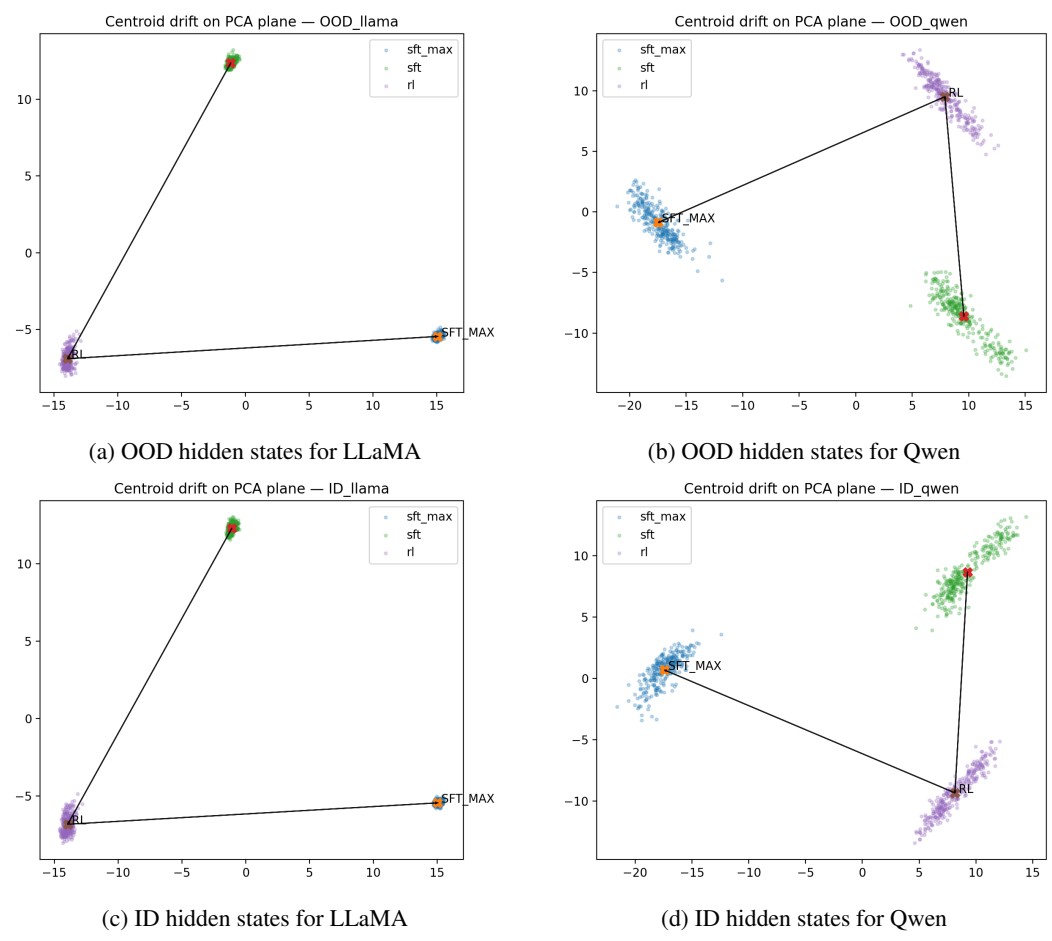

(a) OOD hidden states for LLaMA

(b) OOD hidden states for Qwen

(c) ID hidden states for LLaMA

(d) ID hidden states for Qwen

Figure 17: PCA visualization of the hidden representations at checkpoints $SFT_{MaxOOD}$, $SFT_{END}$ and $RL_{END}$.

We use 300 in-distribution prompts and 300 out-of-distribution prompts to activate hidden states respectively at certain fine-tuning checkpoint, compute PCA for the representation matrix and use the first two principle components to visualize the embedding shifts for both models. We find that RL fine-tuning slightly drags the hidden representation away from the $SFT_{MaxOOD}$, $i.e.,$the embedding distance between $RL_{END}$ and $SFT_{MaxOOD}$ is farther than the $SFT_{END}$ and $SFT_{MaxOOD}$. The representation shift

for Qwen is smaller than LLaMA. This also indicates Qwen is a more robust model than LLaMA during SFT and RL fine-tuning.

### C.11 POTENTIAL OOD FORGETTING EXPLANATION

**Setting**   We do SFT with cross-entropy (CE) on the train set. We evaluate OOD on 24 problems. For each problem we sample up to 6 attempts. Loss is CE on train tokens. OOD accuracy is pass@6 over the 24 OOD problems.

**Verifier and check order (per step)**   1. Format parse: if parse fails → ILLEGAL_FORMAT. Stop other checks for that step.
2. Number check: if numbers in formula are invalid (not from set, wrong count, etc.) → INCORRECT_NUMBER.
3. Solution check: if no valid "final answer" after format checking → NO_SOLUTION.
4. Aggregation: if ≥ 2 of the above are true for a step → also count AGGREGATED_ERR.

**How we compute metrics**   - Loss: mean token-level CE on train data (OOD).
- CORRECT_SOLUTION(CS): a problem is correct if any of its 6 attempts ends with a correct final answer; accuracy is the fraction over 24.
- Step-level rates (NO_SOLUTION(NS), ILLEGAL_FORMAT(IF), INCORRECT_NUMBER(IN), AGGREGATED_ERR(AE)): count steps with the label divided by all steps. Rates can co-occur and do not sum to 1.

| ck | CS | NS | IF | IN | AR | Loss |
|----|----|----|----|----|----|------|
| 10 | 0 | 0.1895 | 0.8006 | 0.0078 | 0.0021 | 1.5411 |
| 20 | 0.0128 | 0.3357 | 0.5462 | 0.0759 | 0.0401 | 0.6293 |
| 30 | 0.1026 | 0.6368 | 0.0099 | 0.2059 | 0.1292 | 0.4262 |
| 40 | 0.094 | 0.8407 | 0.0023 | 0.0713 | 0.069 | 0.5994 |
| 50 | 0.1239 | 0.9109 | 0.0008 | 0.043 | 0.023 | 0.6944 |
| 60 | 0.1624 | 0.9095 | 0.0008 | 0.037 | 0.0228 | 0.7076 |
| 70 | 0.1624 | 0.8644 | 0 | 0.0534 | 0.0527 | 0.7211 |
| 80 | 0.1325 | 0.908 | 0 | 0.0449 | 0.0232 | 0.7273 |
| 90 | 0.141 | 0.8264 | 0 | 0.0797 | 0.068 | 0.7212 |
| 100 | 0.1709 | 0.8776 | 0 | 0.0474 | 0.0434 | 0.7166 |
| 110 | 0.1538 | 0.8854 | 0 | 0.0569 | 0.0292 | 0.7267 |
| 120 | 0.1538 | 0.9182 | 0 | 0.0417 | 0.0118 | 0.7350 |
| 130 | 0.1581 | 0.9066 | 0 | 0.0314 | 0.033 | 0.7625 |
| 140 | 0.1752 | 0.9013 | 0 | 0.0382 | 0.0287 | 0.7660 |
| 150 | 0.1496 | 0.9255 | 0 | 0.029 | 0.018 | 0.7581 |

Table 3: OOD accuracy (trajectory-level pass@6 on 24 problems), step-level error rates, and train CE loss.

**Observation**   - Loss is lowest at ck=30 (0.426) and later rises.
- CORRECT (OOD) grows from 0.00 (ck=10) to ≈ 0.17 (ck=100–150).
- ILLEGAL_FORMAT drops to 0 by ck≥70 (better syntax).
- INCORRECT_NUMBER falls after peaking near ck=30 (better number use).
- AGGREGATED_ERR stays low and trends down.

**Why loss and OOD move differently**   - Different targets: CE fits train tokens; CORRECT measures end-to-end success on OOD with a verifier and pass@6 with verifier feedback.
- Structure over tokens: cleaner format and number use can boost pass@6 even if CE rises.
- Search effect: as more attempts are valid, at-least-one-success increases.
- Apparent "forgetting": later checkpoints may drift from train token distribution (higher CE) while generalizing structure better on OOD (higher CORRECT).

### C.12 FULL RESULTS OF ADVANTAGE DISTRIBUTION

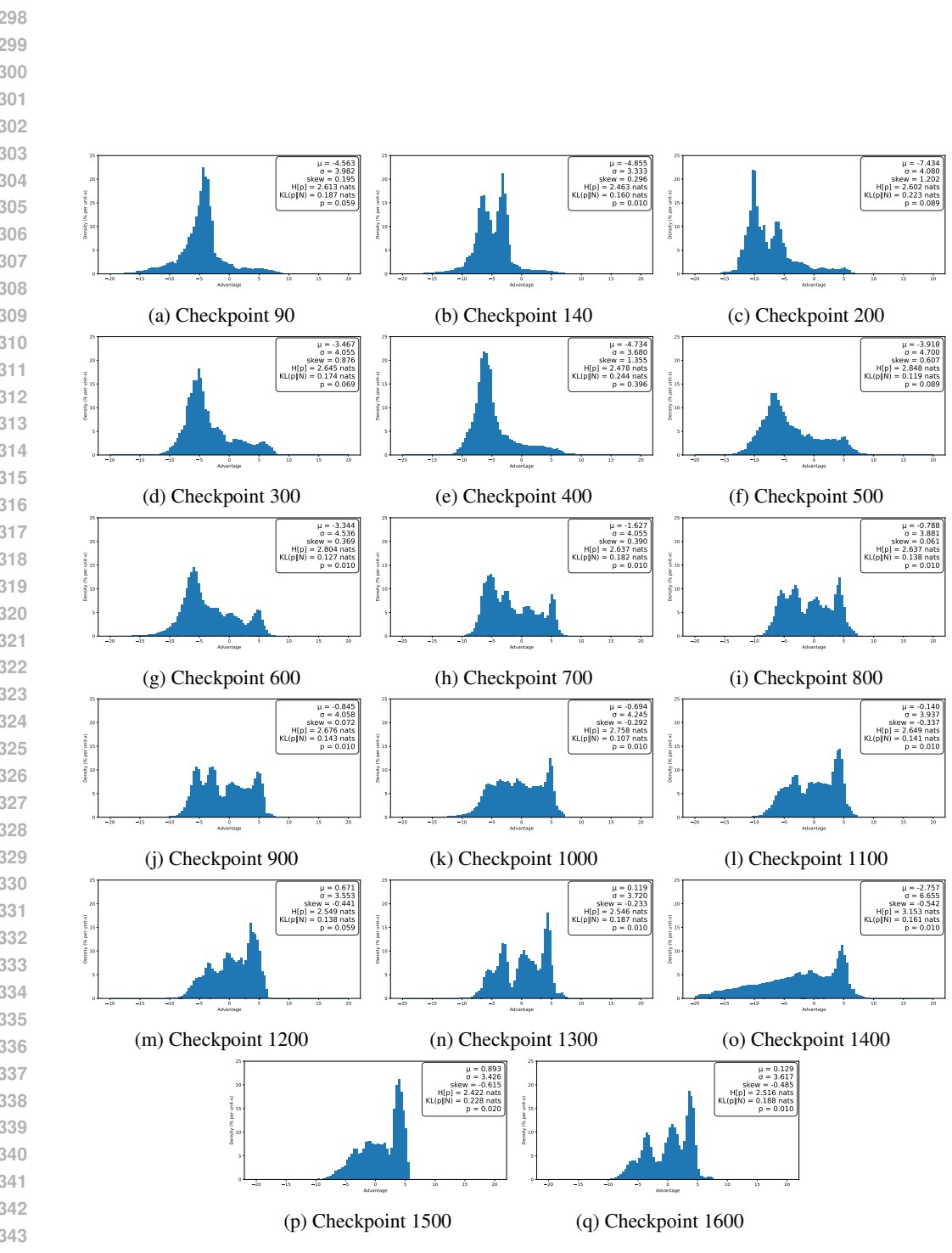

(a) Checkpoint 90     (b) Checkpoint 140     (c) Checkpoint 200

(d) Checkpoint 300     (e) Checkpoint 400     (f) Checkpoint 500

(g) Checkpoint 600     (h) Checkpoint 700     (i) Checkpoint 800

(j) Checkpoint 900     (k) Checkpoint 1000     (l) Checkpoint 1100

(m) Checkpoint 1200     (n) Checkpoint 1300     (o) Checkpoint 1400

(p) Checkpoint 1500     (q) Checkpoint 1600

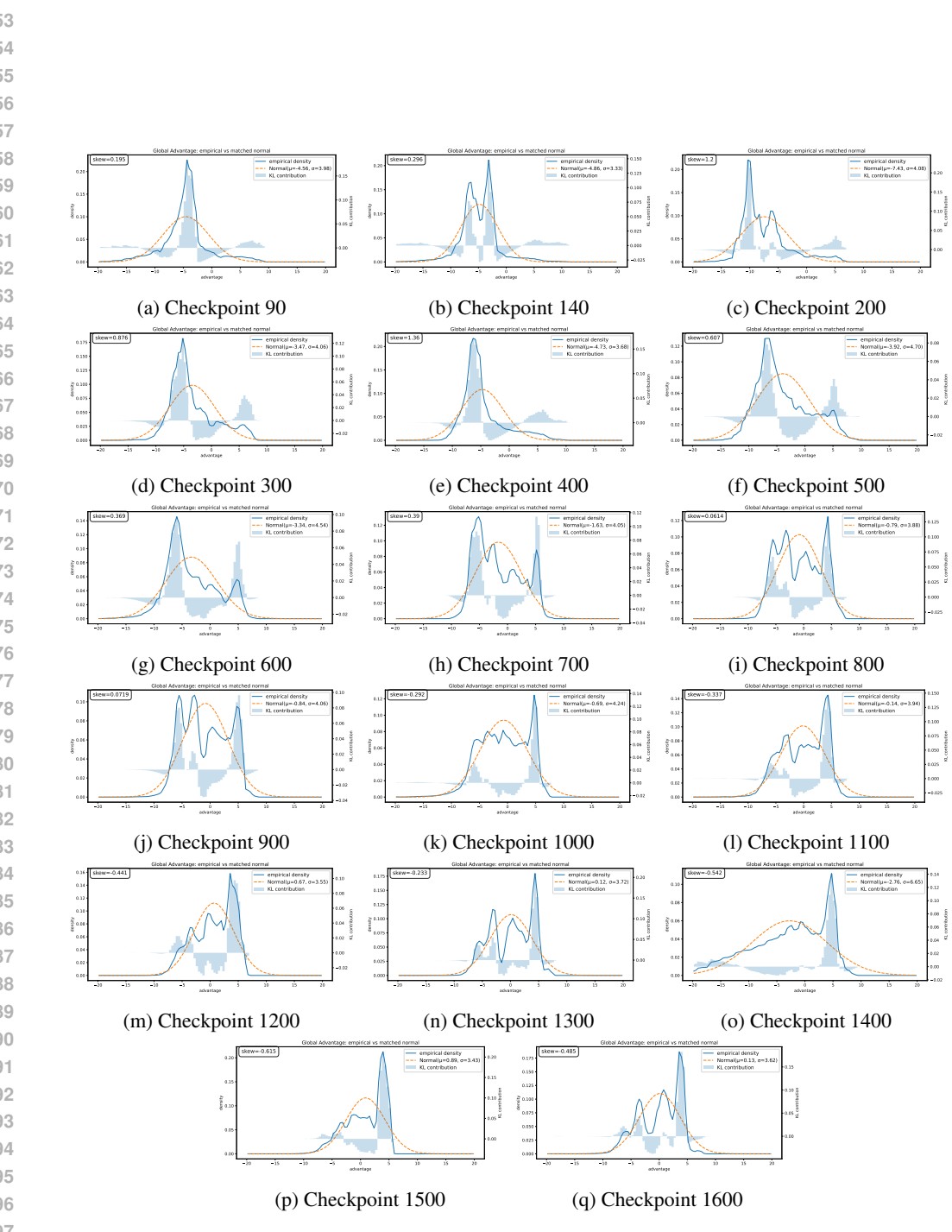

(a) Checkpoint 90  (b) Checkpoint 140  (c) Checkpoint 200

(d) Checkpoint 300  (e) Checkpoint 400  (f) Checkpoint 500

(g) Checkpoint 600  (h) Checkpoint 700  (i) Checkpoint 800

(j) Checkpoint 900  (k) Checkpoint 1000  (l) Checkpoint 1100

(m) Checkpoint 1200  (n) Checkpoint 1300  (o) Checkpoint 1400

(p) Checkpoint 1500  (q) Checkpoint 1600

