# OpenReview forum: "RL Fine-Tuning Heals OOD Forgetting in SFT"
_ICLR.cc/2026/Conference — Submitted to ICLR 2026_

### Official Review · Reviewer_noEb · 2025-10-30

**Soundness:** 3
**Presentation:** 1
**Contribution:** 2
**Rating:** 6
**Confidence:** 4

**Summary:**

This study challenges the oversimplified claim that "SFT memorizes, RL generalizes" by meticulously tracking the OOD reasoning performance of models like LLaMA-3.2-11B and Qwen-2.5-7B throughout full-parameter fine-tuning. The key finding is that OOD performance actually peaks at an early SFT checkpoint and then declines, a phenomenon termed "OOD forgetting", while subsequent RL fine-tuning does not create new generalization ability but instead acts as an automatic recovery mechanism, restoring the lost OOD capability within a specific boundary of SFT training. Through SVD analysis, the paper further reveals that this forgetting and recovery process is primarily driven by the rotation of singular vectors in parameter matrices, not changes in singular values, offering a novel mechanistic insight into the synergy of SFT and RL.

**Strengths:**

1. Through extensive experimentation and analysis from an SVD perspective, this study reveals novel conclusions regarding the effects of SFT and RL on both ID performance and OOD generalization.

2. The experiments and analysis presented in the paper are internally consistent, and the conclusions drawn align well with intuitive expectations, rather than being entirely counterintuitive.

**Weaknesses:**

1. A very poor presentation. The paper extensively uses vspace, resulting in overly crowded content. Additionally, there are numerous errors in the text, such as referring to Qwen-2.5-7B in many places as Qwen-2.5-8B.

2. Regarding Figure 2c, the authors argue that "SFT forgets, RL recovers." However, a closer examination of the curve reveals a more nuanced picture: initially, RL actually performs worse on the OOD task. It is only when the OOD performance of SFT declines significantly that RL begins to demonstrate a recovery effect at intermediate checkpoints. Moreover, in the final stages of training, RL once again starts to impair OOD performance. The authors do not fully explain these dynamic patterns observed in the curve.

3. I still believe that conducting ID and OOD analysis solely on the GeneralPoints task is insufficient. Conclusions drawn from a single task are highly susceptible to misinterpretation. For example, the conclusion that "SFT memorizes, while RL generalizes" was also derived from the GeneralPoints task, yet it contradicts the findings presented in this paper. Therefore, relying on experiments from just one task may lead to unreliable conclusions, as they could be influenced by external factors such as the base model, learning rate, and others.

**Questions:**

Regarding Figure 2d, why does RL consistently underperform SFT on the ID task throughout the entire process? Could this be due to inadequate RL training in the paper's experiments, resulting in poorer model performance rather than an inherent characteristic?

---

> ### Author Response · Authors · 2025-11-21
> **Response to Reviewer noEb**
>
> ### $\textbf{W1: A very poor presentation.}$
> The paper extensively uses vspace, resulting in overly crowded content. Additionally, there are numerous errors in the text, such as referring to Qwen-2.5-7B in many places as Qwen-2.5-8B.
> ### $\textbf{Reply 1:}$
> Thanks for your constructive feedback on our presentation. We will 1) try our best to simplify our expression so that it does not look too crowded; 2) fix the typos accordingly.
> ### $\textbf{W2: }$
> Regarding Figure 2c, the authors argue that "SFT forgets, RL recovers." However, a closer examination of the curve reveals a more nuanced picture: initially, RL actually performs worse on the OOD task. It is only when the OOD performance of SFT declines significantly that RL begins to demonstrate a recovery effect at intermediate checkpoints. Moreover, in the final stages of training, RL once again starts to impair OOD performance. The authors do not fully explain these dynamic patterns observed in the curve.
> ### $\textbf{Reply 2: }$
> Thanks for your careful reading on Figure 2c. Your summary of Figure 2c is very accurate, comprehensive and is consistent with what we have analyzed in section 3.2. The reason that we use the simplified argument "SFT forgets, RL recovers" is to contrast with the argument "SFT memorizes, RL generalizes" in [1], therefore we follow the same format. But in details, Figure 2c indeed shows much richer information than [1], e.g. 1) The recovery of RL on OOD forgetting is only effective within a range of SFT checkpoints; 2) The recovered OOD performance barely surpasses the peak of SFT, which means RL does not fundamentally improve the OOD reasoning ability.
> ### $\textbf{W3: }$
> I still believe that conducting ID and OOD analysis solely on the GeneralPoints task is insufficient. Conclusions drawn from a single task are highly susceptible to misinterpretation. For example, the conclusion that "SFT memorizes, while RL generalizes" was also derived from the GeneralPoints task, yet it contradicts the findings presented in this paper. Therefore, relying on experiments from just one task may lead to unreliable conclusions, as they could be influenced by external factors such as the base model, learning rate, and others.
> ### $\textbf{Reply 3:}$
> We agree with your belief that a conclusion should not be made only based on a single task. And this is why we also verify our claim "SFT forgets, RL recovers" on $\textit{{Navigation}}$ and $\textit{{Rank-Determinant Computation}}$, which additionally evaluate the spatial and cross-concept math reasoning abilities of models. The results are shown in Figure 1, demonstrating that RL can heal the OOD forgetting at checkpoint SFT-end but barely surpasses the best of SFT.
>
> Actually, we have mentioned in our paper that the conclusion "SFT memorizes, RL generalizes" from [1] does not contradict our findings, as shown in the comparison of SFT-end and RL-end in Figure 1. Their limitation is that they did not discover the SFT-MaxOOD checkpoint, which means they only summarize the partial relation between SFT and RL with a single SFT checkpoint and lead to oversimplified conclusion. In addition, we evaluate the models on more diverse OOD benchmarks accordingly (arc_c, commonsenseqa, gpqa, ifeval_loose, mmlu_pro). And in 9/10 cases (except Qwen on ifeval_loose_test), the results align with our observations that the RL heals the OOD forgetting in SFT but barely surpasses the best of SFT.
>
> | Model | Tasks\Checkpoints          | Max_OOD | SFT_End    | RL_End     |
> | ----- | ------------------ | ------- | ------ | ------ |
> | Llama | arc_c_test         | 0.8     | 0.7    | 0.8    |
> |       | commonsenseqa_test | 0.81  | 0.62 | 0.69 |
> |       | gpqa_test          | 0.44    | 0.32   | 0.44   |
> |       | ifeval_loose_test  | 0.27  | 0.27 | 0.27 |
> |       | mmlu_pro           | 0.46  | 0.36 | 0.37 |
> | Qwen  | arc_c_test         | 0.92    | 0.88   | 0.88   |
> |       | commonsenseqa_test | 0.79  | 0.73   | 0.75   |
> |       | gpqa_test          | 0.44     | 0.42   | 0.44   |
> |       | ifeval_loose_test  | 0.63 | 0.67 | 0.7    |
> |       | mmlu_pro           | 0.73  | 0.66 | 0.66 |
> ### $\textbf{W4:}$
> Regarding Figure 2d, why does RL consistently underperform SFT on the ID task throughout the entire process? Could this be due to inadequate RL training in the paper's experiments, resulting in poorer model performance rather than an inherent characteristic?
> ### $\textbf{Reply 4:}$
> In our RL training, we follow the same setting as [1] and use 4XH100 to train 24 hours or 8XH100 to train 12 hours depending on the availability of resources. Continuing RL training will lead to further decline for both ID and OOD performance. Such model deterioration of long-term RL training was also found by other researchers or practitioners. Therefore, we believe that we don't have inadequate training issue in our RL settings.
>
> [1] Tianzhe Chu, et al. Sft memorizes, rl generalizes: A comparative study of foundation model post-training, 2025.

---

### Official Review · Reviewer_vU7W · 2025-11-03

**Soundness:** 3
**Presentation:** 2
**Contribution:** 2
**Rating:** 4
**Confidence:** 4

**Summary:**

The paper studies reinforcement-learning fine-tuning of LLMs and argues that RLFT is surprisingly effective at improving out-of-distribution generalization.
The experiments show that OOD performance actually peaks early during SFT and then declines, a phenomenon they call "OOD forgetting." The subsequent RL stage doesn't create fundamentally new reasoning abilities but instead acts as a "healer," recovering the OOD capabilities that were lost during the later parts of SFT.

**Strengths:**

- Highlights the value of RL even without explicitly OOD data.
- The authors go beyond reporting gains, offering plausible hypotheses about why RLFT helps.

**Weaknesses:**

- The OOD benchmarks resemble the ID data distribution; thus, it’s unclear how well the claims hold under truly challenging shifts. Stronger or more diverse OOD settings are needed.
- The paper states that PPO was used rather than GRPO. What is the rationale behind choosing PPO to train the additional critic model, given that most recent RL works use GRPO?
- The high-level insight that RLFT can mitigate OOD degradation, is intuitively expected and not particularly surprising for the community.
- Several plots (e.g., Fig. 2(a)(b), Fig. 3) are difficult to interpret without reading the text carefully.
- It is unclear why advantage estimation is necessary for mitigating OOD forgetting, especially if the "restorability" of checkpoints can be reasonably inferred from IID accuracy.
- The paper reports RL-End having better accuracy than SFT-End in Fig. 1 but underperforming SFT-End in Fig. 2c. The authors should clarify such inconsistency.
- A more appropriate comparison would be: SFT(ID) + RL vs. SFT(ID) + SFT(OOD) to show RL benefits over fine-tuning on OOD data.
- Experiments are restricted to a narrow set of base models and OOD tasks, making it unclear whether the conclusions generalize.

**Questions:**

See weaknesses.

---

> ### Author Response · Authors · 2025-11-21
> **Response to Reviewer vU7W Part 1/2**
>
> ### $\textbf{W1:}$
> The OOD benchmarks resemble the ID data distribution; thus, it’s unclear how well the claims hold under truly challenging shifts. Stronger or more diverse OOD settings are needed.
>
> ### $\textbf{Reply 1:}$
>
> Thank you for your suggestions. For $\textit{GeneralPoints}$ and $\textit{Navigation}$, we follow the same OOD setting as [1]. For $\textit{Rank-Determinant Computation}$, the training and test datasets are two different math tasks. In addition, we evaluate the models on more diverse OOD benchmarks accordingly (arc_c, commonsenseqa, gpqa, ifeval_loose, mmlu_pro). And in 9/10 cases (except Qwen on ifeval_loose_test), the results align with our observations that the RL heals the OOD forgetting in SFT but barely surpasses the best of SFT.
>
> | Model | Tasks\OOD_Eval_Checkpoints          | Max_OOD | SFT_End    | RL_End     |
> | ----- | ------------------ | ------- | ------ | ------ |
> | **Llama** | arc_c_test         | 0.8     | 0.7    | 0.8    |
> |       | commonsenseqa_test | 0.8077  | 0.6154 | 0.6923 |
> |       | gpqa_test          | 0.44    | 0.32   | 0.44   |
> |       | ifeval_loose_test  | 0.2667  | 0.2667 | 0.2667 |
> |       | mmlu_pro           | 0.4571  | 0.3571 | 0.3714 |
> | **Qwen**  | arc_c_test         | 0.92    | 0.88   | 0.88   |
> |       | commonsenseqa_test | 0.7885  | 0.73   | 0.75   |
> |       | gpqa_test          | 0.44     | 0.42   | 0.44   |
> |       | ifeval_loose_test  | 0.63333 | 0.6667 | 0.7    |
> |       | mmlu_pro           | 0.7286  | 0.6571 | 0.6571 |
>
> ### $\textbf{W2:}$
> The paper states that PPO was used rather than GRPO. What is the rationale behind choosing PPO to train the additional critic model, given that most recent RL works use GRPO?
>
> ### $\textbf{Reply 2:}$
>
> We follow the setup in [1] to make a fair comparison with their results, where they use PPO to update the policy parameters. In addition, the popular algorithms like GRPO that you mentioned is an efficient and computational friendly PPO-like framework. Therefore, We find that PPO (which is used in "SFT memorizes, RL generalizes") is a more representative method for the analysis. Furthermore, although the GRPO saves a critic network, it causes an OOM issue when we use a large group_size like 8 or 16 during training. But as you suggested, we have resubmitted the GRPO (group_size=4) with the GeneralPoints task to the queue; we will update the results as soon as they are available.
>
> ### Update: GRPO results
> | **Model** | **Dataset** | **MAX_OOD** | **SFT_END** | **RL** |
> | --------- | ----------- | ----------- | ----------- | ------ |
> | **Llama** | OOD         | 0.1752      | 0.0692      | 0.1325 |
> |           | ID          | 0.2479      | 0.6838      | 0.735  |
> | **Qwen**  | OOD         | 0.1967      | 0.1709      | 0.1795 |
> |           | ID          | 0.2607      | 0.4359      | 0.4573 |
>
> The table above is the GRPO results of both models for task GeneralPoints. Interestingly, GRPO performs better than PPO in terms of In-distribution and worse than PPO for OOD; overall, both algorithms aligns with our claim that RL heals OOD forgetting but does not surpass the best of SFT.
>
>
> ### $\textbf{W3:}$
> The high-level insight that RLFT can mitigate OOD degradation, is intuitively expected and not particularly surprising for the community.
>
> ### $\textbf{Reply 3:}$
>
> Thanks for bringing this discussion on the table. For the community, the uniqueness of our findings are: 1) $\textbf{RLFT does not always mitigate OOD degradation}$. Based on the whole evolution process that we depict about the synergy of RL and SFT, there exists an effective boundary of the SFT checkpoints that RL can recover the OOD degradation. And when outside the boundary, RLFT will worsen the OOD degradation; 2)  The recovered OOD performance barely surpasses the best of SFT, which is lacking in the community. We depict the effectiveness of RLFT more exactly and comprehensively, beyond just intuition.
>
> Besides, we are aware of some recent papers that study the relevant topic about RLFT and forgetting or OOD degradation [2,3,4] (although they were released near or after ICLR submission deadline). However, their claims are either in continual learning setting or trying to compare RL vs. SFT instead of the two-stage SFT+RL finetuning. In our paper,
>
> - We not only provide an intuition about the mitigation of RLFT on OOD degradation, but also describe the whole evolution of the synergy of RL and SFT.
>
> - We do not study the degradation in a continual learning setting. In other words, we do not test the fine-tuned models on previously trained tasks, but on untrained tasks.
>
> ### $\textbf{W4:}$
>  Several plots (e.g., Fig. 2(a)(b), Fig. 3) are difficult to interpret without reading the text carefully.
>
> ### $\textbf{Reply 4:}$
>
> Thanks for your feedback. Due to space limitations, we have to zoom out the figures. We will try to make them as clear as possible in the revised version.

---

> ### Author Response · Authors · 2025-11-21
> **Response to Reviewer vU7W Part 2/2**
>
> ### $\textbf{W5:}$
>  It is unclear why advantage estimation is necessary for mitigating OOD forgetting, especially if the "restorability" of checkpoints can be reasonably inferred from IID accuracy.
>
> ### $\textbf{Reply 5:}$
>
> We would like to clarify the motivation and logic behind this part of the analysis.
>
> For the analysis of advantage distribution, our motivation is trying to figure out why the mitigation effects of RL are different between checkpoints inside and outside the boundary. Based on equation (3), the most straightforward way is to check out the differences of their advantage estimation distributions and see if you could find out any useful pattern. Therefore, the logic here is not to claim that advantage estimation is necessary for mitigating OOD forgetting. It is first to explore the factors that impact the behaviors of RL on OOD forgetting, and the advantage estimation happens to be one of the most important properties that we need to explore.
>
> For ID accuracy, since the RLFT is the subsequent step after SFT, thus, the beginning of RL is actually the end of SFT. So by observing the ID accuracy of SFT, we can approximately infer the proportion of positive/negative rewards that it receives at the early RL stage. And we find that in the effective RL boundary, such a proportion suggests a balanced fraction of positive/negative. The logic here is not to infer "restorability" of checkpoints from ID accuracy of SFT. It is to infer the RL reward distribution from ID accuracy to study the "restorability".
>
>
> ### $\textbf{W6:}$
>
> The paper reports RL-End having better accuracy than SFT-End in Fig. 1 but underperforming SFT-End in Fig. 2c. The authors should clarify such an inconsistency.
>
> ### $\textbf{Reply 6:}$
>
> We don't have SFT-End in Figure 2c. In legend, the orange line is for SFT+RL. As mentioned in our paper, the specific checkpoints SFT-End and RL-End in Figure 1 correspond to the results at checkpoint 1100 in Figure 2c (indicated as SFT, SFT+RL), and their results are consistent.
>
>
> ### $\textbf{W7: }$
>
> A more appropriate comparison would be: SFT(ID) + RL vs. SFT(ID) + SFT(OOD) to show RL benefits over fine-tuning on OOD data.
>
> ### $\textbf{Reply 7:}$
>
> We generate 5k OOD data(GeneralPoints) and mix it into the SFT training dataset. After supervised fine-tuning, the In-distribution accuracy is lower than the SFT(ID) setting around 5-10%, but the OOD accuracy is much higher than the SFT(ID) setting around 20-30%. This is because the SFT(ID) + SFT(OOD) = new SFT(ID), the OOD becomes a new ID dataset, but the forgetting still exists in other unseen tasks, such as commonsenseqa which we evaluated in W1. In this case, we still claim that RL is helpful because we cannot collect the entire OOD dataset for SFT in the real world, and we need RL to explore more diverse OOD tasks.
>
> ### $\textbf{W8:}$
>
> Experiments are restricted to a narrow set of base models and OOD tasks, making it unclear whether the conclusions generalize.
>
> ### $\textbf{Reply 8:}$
>
> We definitely agree that we should use more models and tasks to make the conclusion more solid. In our paper, We adopted two most popular open-source models (Llama and Qwen) to verify our claim on three diverse tasks. In addition, we tested on another five popular OOD datasets and re-evaluate the trained models. The results are shown in Reply 1 and the conclusion is verified to be generalizable.
>
> Additionally, due to the limited computational resources and the high expenses on the experiments with different models, checkpoints and datasets for the academic community (e.g., in the RL stage, we need to use 4XH100 to train for 24 hours or 8XH100 to train for 12 hours for each single setting of [model, checkpoint, task]), we have already tried our best to verify our claims on as many and diverse tasks as possible. Thank you for your suggestion and understanding.
>
> [1] Tianzhe Chu, et al. Sft memorizes, rl generalizes: A comparative study of foundation model post-training, ICML 2025.
>
> [2] Lai, S., et al. (2025). Reinforcement fine-tuning naturally mitigates forgetting in continual post-training. arXiv preprint arXiv:2507.05386.
>
> [3] Shenfeld, I., et al. (2025). RL's Razor: Why Online Reinforcement Learning Forgets Less. arXiv preprint arXiv:2509.04259.
>
> [4] Yuan, X., et al. (2025). Mitigating Forgetting Between Supervised and Reinforcement Learning Yields Stronger Reasoners. arXiv preprint arXiv:2510.04454.

---

### Official Review · Reviewer_hdfn · 2025-11-06

**Soundness:** 3
**Presentation:** 2
**Contribution:** 3
**Rating:** 6
**Confidence:** 4

**Summary:**

The paper asks an important question: how does SFT interact with a consecutive RL training?
This two-step post training procedure has played major role in the LLM era and yet the relationship between the two steps is poorly understood.
The authors claim that SFT initially improves OOD performance and then there's a drastic degradation in OOD performance as SFT continues. RL salvages some of that, but interestingly enough, we could find a checkpoint earlier in SFT that does better.
By analyzing many checkpoints of SFT training and running RL on them, the authors show at which points in SFT can we come in and do RL to improve performance, identifying "boundaries" in which RL can take effect. The authors then explain the results by analyzing the advantage distributions from PPO empirically, and then perform an SVD analysis on weight matrices of different checkpoints to show that most of the effects observed in post-training come from rotation of the singular vectors, with the singular values roughly fixed.

**Strengths:**

Important empirical findings: The paper provides valuable insights into the SFT-RL dynamics, showing that the best OOD performance occurs at intermediate SFT checkpoints rather than at convergence. This has immediate practical implications for practitioners.

Comprehensive experimental design: The continuous tracking of performance across multiple checkpoints and the use of controlled reasoning tasks provides solid evidence for the claims.

Novel mechanistic insights: The SVD analysis revealing that singular vectors rotate while values remain stable is genuinely interesting and challenges existing assumptions about how fine-tuning affects model capacity.

Practical boundaries identified: The discovery that RL recovery only works within certain SFT checkpoint ranges (roughly 40-80% ID accuracy) and the connection to advantage distribution properties provides actionable guidance.

**Weaknesses:**

Poor introduction organization: The paper dives into technical details without clearly stating contributions upfront. The introduction mentions "in Section 6 we will conclude" without summarizing what happens in between, making it hard to follow the narrative arc.

Presentation issues:
Key concepts like "OOD forgetting" need clearer, earlier definition
Figure 2 has inconsistent axes labels
Figure 3 plots are impossible to read in detail
The 0.4-0.8 accuracy claim for RL effectiveness should be presented as a percentage (I think)

Insufficient experimental support:

More random seeds needed to substantiate the "clear boundaries" claim (I found that finetuning evals are noisy and it's important to explore things more broadly).
The connection between Gaussian metrics and histograms isn't well explained
"Negative rewards on examples outside the training set" needs clarification - does this mean high perplexity?


Weak mechanistic claims:

The "RL as Automatic Mitigation" section is handwavy - dropout and weight decay effects aren't studied here despite being cited as analogies
The claim about "saving repetition work" is vague
The paper asks important questions about SFT-RL interaction but doesn't fully answer them

**Questions:**

Run additional seeds for the boundary experiments to strengthen statistical claims
Clean up Figure 3
Remove statements on weight decay, dropout, saving repetition work, etc.

---

> ### Author Response · Authors · 2025-11-21
> **Response to Reviewer hdfn Part 1/2**
>
> We appreciate your constructive feedback and here are our replies to your concerns and questions.
>
> ### $\textbf{W1: Poor introduction organization}$
> The paper dives into technical details without clearly stating contributions upfront. The introduction mentions "in Section 6 we will conclude" without summarizing what happens in between, making it hard to follow the narrative arc.
>
> ### $\textbf{Reply 1:}$
> Thanks for your suggestions on the "introduction organization" issue. To fix it, in the introduction section, we will highlight 1) our observation of OOD peak, OOD forgetting and OOD recovery behavior in the two-stage fine-tuning process, and our differences with some other RL reasoning papers focused on the in-distribution generalization abilities of the fine-tuned models; 2) the differences of our OOD setting and traditional forgetting issues in continual learning, where they focus on the forgetting in previously trained tasks, but we focus on more general reasoning abilities which are not trained before; 3) RL can recover the OOD forgetting in SFT, but only within an effective boundary, which is a novel discovery.
>
> In the conclusion section, we will provide more details and emphasize the connection between each section accordingly.
>
> ### $\textbf{W2: Presentation issues}$
> Key concepts like "OOD forgetting" need a clearer, earlier definition. Figure 2 has inconsistent axes labels. Figure 3 plots are impossible to read in detail. The 0.4-0.8 accuracy claim for RL effectiveness should be presented as a percentage (I think)
>
> ### $\textbf{Reply 2:}$
>
> Thanks for your suggestion.
> - We have mentioned OOD forgetting in the introduction section and it refers to "OOD reasoning performance will peak rapidly in very early stage of SFT and then degrades slowly as SFT continues". We will highlight this definition and make it clearer accordingly.
>
> - In Figure 2, different subfigures show different aspects of the evaluation of the fine-tuning performance. Therefore, their y-axis are different. Subfigure (a) shows the ID training/test loss, which measures the token-wise error against the supervision; Subfigure (b) shows the proportion of generated answers that can correctly follow the right format; Subfigure (c,d) measure the proportion of generated answers which can correctly solve the task in in-distribution and out-of-distribution settings.
>
> - Due to space limitations, we will update Figure 3 later to make it more reader-friendly. Thanks for your feedback.
>
> - We will take your advice and use percentage for the accuracy.

---

> ### Author Response · Authors · 2025-11-21
> **Response to Reviewer hdfn Part 2/2**
>
> ### $\textbf{W3: Insufficient experimental support}$
> More random seeds needed to substantiate the "clear boundaries" claim (I found that finetuning evals are noisy and it's important to explore things more broadly). The connection between Gaussian metrics and histograms isn't well explained. "Negative rewards on examples outside the training set" needs clarification - does this mean high perplexity?
>
> ### $\textbf{Reply 3:}$
> Thanks for your feedback.
> - Due to the limited computational resources in our academic institution (e.g., during the RL stage, we need to use 4 x H100 to train for 24 hours or 8 x H100 to train for 12 hours for each single setting of [model, checkpoint, task, seed], and the cluster has been very crowded), we have tried our best to verify our claims with more random seeds as follows: we selected three representative SFT checkpoints (400, 900, and 1600 on GeneralPoints) and do RL fine-tuning with different seeds; we found that only the OOD accuracy at checkpoint 900 can be recovered by RL, which aligns with our claim. The differences between the results of different seeds are around 1.5%. We also adopted different seeds to evaluate the model; the variance is negligible.
>
> - Gaussian metrics and histograms: The histograms are the advantage estimation distributions in PPO at different checkpoints, which are the most critical factor to control the evolution of RL. We try to study the differences between them at the checkpoints which are inside and outside the effective boundary of RL to understand how they impact RL. The Gaussian metric is a measurement to study one aspect of their properties, i.e, non-Gaussianity. We will add more explanation and clarify it accordingly.
>
> - Does "Negative rewards on examples outside the training set" mean high perplexity:
> They have some similar places, but not exactly the same. We will clarify it as follows.
> In SFT, we only reward the single reference answer in the training data. And other outputs, including those that are mathematically correct formulas but not in the training set, are treated just like wrong answers. In other words, their probabilities are pushed down by the gradient. From an RL perspective, that is like giving a positive reward to the reference sequence and an implicit negative reward to other sequences. Therefore, when the model generates a correct but unseen solution (outside the training set), the cross-entropy loss gives gradients that decrease its probability. This can lead to high perplexity on those newly-explored correct solutions. We will clarify this paragraph accordingly.
>
> ### $\textbf{W4: Weak mechanistic claims}$
>
> The "RL as Automatic Mitigation" section is handwavy - dropout and weight decay effects aren't studied here despite being cited as analogies. The claim about "saving repetition work" is vague.
>
> The paper asks important questions about SFT-RL interaction but doesn't fully answer them
>
> ### $\textbf{Reply 4:}$
>
> - We will remove the "RL as Automatic Mitigation" paragraph to avoid confusion in your reading. What we would like to convey in this paragraph is, although RL finetuned model barely surpasses the peak of SFT, it dies not mean that RL is useless for post-training. Because in practice, it is impossible to check out each SFT checkpoint repeatedly and pick up the best one due to the computational costs. Instead, we can simply select a reasonable SFT checkpoint, and then use RL and still get competitive results. Therefore, this saves us lots of manual work and still can get good but not the best OOD performance.
>
> - SFT-RL interaction: Our observation on the interactions between SFT and RL is mentioned in our paper. We will summarize it here for you: SFT first improves and then starts to hurt OOD ability at certain checkpoints, i.e., OOD forgetting; RL can heals the lost OOD ability in SFT within a boundary of SFT checkpoints, where RL can receive richer and balanced positive/negative reward signals; However, SFT+RL barely surpasses the peak of SFT, which means RL can only partially recover the damaged OOD ability and cannot foundamentally escape the limitation of the base model.

---

### Official Review · Reviewer_Bxpq · 2025-11-08

**Soundness:** 2
**Presentation:** 2
**Contribution:** 2
**Rating:** 2
**Confidence:** 3

**Summary:**

The paper studies SFT and RL dynamics on three small tasks and claims: (1) OOD performance peaks at an early SFT checkpoint and then declines; (2) RL can recover some OOD from late SFT but does not surpass the SFT peak; (3) there is a narrow “RL window” where starting RL from mid-SFT helps; and (4) an SVD analysis says rotations of singular vectors (U,V), rather than changes in singular values, track OOD forgetting/recovery.

**Strengths:**

- The paper focuses on understanding how SFT and RL affect OOD generalization, a timely and practically important question for current training pipelines.


- The snapshot view comparing $SFT_{MaxOOD}$, $SFT_{End}$, and $RL_{End}$ is straightforward and helps readers quickly see the trade-off between ID gains and OOD degradation.


- The checkpoint-sweep experiment effectively illustrates how OOD performance evolves during SFT and how RL recovers (or fails to recover) performance depending on the starting checkpoint.

**Weaknesses:**

`W1: Limited novelty of the claimed insight`

The observation that the best OOD performance occurs at some intermediate SFT checkpoint is already implied by prior work. It is shown clearly, for example, in Figure 1 of “SFT Memorizes, RL Generalizes” (https://arxiv.org/abs/2501.17161), where certain tasks exhibit exactly this behavior. Reframing this as a new contribution feels redundant and adds little conceptual novelty.

`W2: Checkpoint-sweep analysis shown for only one task`

The key dynamic plots (ID/OOD accuracy vs. SFT checkpoint, before and after RL) are presented only for the GeneralPoints task on LLaMA. The other two tasks—Navigation and Matrix—are not analyzed in the same way, making it unclear whether the reported trends generalize beyond this single example.

`W3: Qwen results do not support the main claim`

In the appendix experiments for Qwen, RL never surpasses the corresponding SFT checkpoint on OOD accuracy and often performs worse on ID. This suggests that the claimed “RL recovery” phenomenon depends heavily on the chosen starting point and is not a consistent or universal trend.

`W4: RL results limited to a single algorithm (PPO)`

All experiments use PPO with fixed hyperparameters, yet the paper generalizes its findings to “RL” broadly. Without testing other widely used algorithms (e.g., GRPO, REINFORCE with baseline, or preference-based methods like DPO), the conclusions remain specific to PPO and its particular optimization behavior.

`W5: Insufficient empirical validation`

The experimental evidence is thin: no error bars or seed variances are reported, and the main claims are demonstrated primarily on one model (LLaMA) and one task (GeneralPoints). This makes it difficult to assess statistical reliability or generality.

`W6: Minor issues`

- W6.1: The paper refers to “LLaMA-3.2-11B,” but such a model does not exist. It likely refers to LLaMA-3.2-11B-Vision, and this should be clarified.

- W6.2: There is no full checkpoint sweep for Qwen on the GeneralPoints task in the main text; only a late-stage subset is shown, where RL underperforms. Including the full curve would make the results more transparent and comparable to LLaMA.

**Questions:**

Please refer to weaknesses.

---

> ### Author Response · Authors · 2025-11-24
> **Response to Reviewer Bxpq Part 1/2**
>
> ### $\textbf{W1: Limited novelty of the claimed insight}$
>
> The observation that the best OOD performance occurs at some intermediate SFT checkpoint is already implied by prior work. It is shown clearly, for example, in Figure 1 of “SFT Memorizes, RL Generalizes” (https://arxiv.org/abs/2501.17161), where certain tasks exhibit exactly this behavior. Reframing this as a new contribution feels redundant and adds little conceptual novelty.
>
> ### $\textbf{Reply 1:}$
>
> Thanks for your feedback and the given reference. However, after carefully reading Figure 1 in “SFT Memorizes, RL Generalizes”, we cannot find any implication made by the authors about "best OOD performance occurs at some intermediate SFT checkpoint". The original content from that paper is "SFT appears to memorize the training rules and does not generalize (see Figure 1 for an example)."
>
> In addition, the observation of OOD forgetting in SFT is only a small part of our contributions and novelty. Besides, we study the role of RL in the two stage SFT+RL paradigm, and have some novel findings that 1) RL can heal the OOD forgetting in SFT, but is only effective in a certain range of checkpoints; 2) the recovered OOD performance does not surpass the peak of SFT.
>
> ### $\textbf{W2: Checkpoint-sweep analysis shown for only one task}$
>
> The key dynamic plots (ID/OOD accuracy vs. SFT checkpoint, before and after RL) are presented only for the GeneralPoints task on LLaMA. The other two tasks—Navigation and Matrix—are not analyzed in the same way, making it unclear whether the reported trends generalize beyond this single example.
>
>
> ### $\textbf{Reply 2:}$
>
> Due to the limited computational resources and the high expenses on the experiments with different models, checkpoints and datasets (e.g., in the RL stage, we need to use 4XH100 to train for 24 hours or 8XH100 to train for 12 hours for each single setting of [model, checkpoint, task]), we cannot finish the analysis on all checkpoints for other tasks. However, we have tried our best to verify our claims on as many and diverse tasks (arc_c, commonsenseqa, gpqa, ifeval_loose, mmlu_pro) as possible. And the results below align with our observations that the RL heals the OOD forgetting in SFT but barely surpasses the best of SFT.
>
> | Model | Tasks\Checkpoints          | Max_OOD | SFT_End    | RL_End     |
> | ----- | ------------------ | ------- | ------ | ------ |
> | Llama | arc_c_test         | 0.8     | 0.7    | 0.8    |
> |       | commonsenseqa_test | 0.81  | 0.62 | 0.69 |
> |       | gpqa_test          | 0.44    | 0.32   | 0.44   |
> |       | ifeval_loose_test  | 0.27  | 0.27 | 0.27 |
> |       | mmlu_pro           | 0.46  | 0.36 | 0.37 |
> | Qwen  | arc_c_test         | 0.92    | 0.88   | 0.88   |
> |       | commonsenseqa_test | 0.79  | 0.73   | 0.75   |
> |       | gpqa_test          | 0.44     | 0.42   | 0.44   |
> |       | ifeval_loose_test  | 0.63 | 0.67 | 0.7    |
> |       | mmlu_pro           | 0.73  | 0.66 | 0.66 |
>
>
> ### $\textbf{W3: Qwen results do not support the main claim}$
>
> In the appendix experiments for Qwen, RL never surpasses the corresponding SFT checkpoint on OOD accuracy and often performs worse on ID. This suggests that the claimed “RL recovery” phenomenon depends heavily on the chosen starting point and is not a consistent or universal trend.
> ### $\textbf{Reply 3:}$
>
> Thanks for your comments. We didn't claim that "RL never surpasses the corresponding SFT checkpoint on OOD accuracy" in Appendix. Actually in Figure 11 in Appendix C.4, we only partially show that, after too many SFT checkpoints (over the upper bound of the effective boundary), RL cannot recover the lost OOD performance, which is consistent with what we have shown in the main paper.
>
> If you would like to find out more results for Qwen, please refer to Figure 1 in our main paper, and the results are consistent with our main conclusions that RL heals the OOD forgetting at some checkpoints in SFT, but does not surpass the peak in SFT.

---

> > ### Comment · Reviewer_Bxpq · 2025-11-25
> >
> > Thank you for your responses.
> >
> > I just realized that in my W1, I accidentally referred to Figure 1. What I intended to cite was Figure 5 in the same paper (https://arxiv.org/pdf/2501.17161). Could you please comment on this?

---

> ### Author Response · Authors · 2025-11-24
> **Response to Reviewer Bxpq Part 2/2**
>
> ### $\textbf{W4: RL results limited to a single algorithm (PPO)}$
>
> All experiments use PPO with fixed hyperparameters, yet the paper generalizes its findings to “RL” broadly. Without testing other widely used algorithms (e.g., GRPO, REINFORCE with baseline, or preference-based methods like DPO), the conclusions remain specific to PPO and its particular optimization behavior.
>
> ### $\textbf{Reply 4:}$
>
> | **Model** | **Dataset** | **MAX_OOD** | **SFT_END** | **RL** |
> | --------- | ----------- | ----------- | ----------- | ------ |
> | **Llama** | OOD         | 0.1752      | 0.0692      | 0.1325 |
> |           | ID          | 0.2479      | 0.6838      | 0.735  |
> | **Qwen**  | OOD         | 0.1967      | 0.1709      | 0.1795 |
> |           | ID          | 0.2607      | 0.4359      | 0.4573 |
>
> Thanks for your suggestion. We've verified our claims with GRPO and the results of both models on GeneralPoints are shown in the above table.  Interestingly, GRPO performs better than PPO in terms of In-distribution and worse than PPO for OOD; overall, both algorithms aligns with our claim that RL heals OOD forgetting but does not surpass the best of SFT.
>
> ### $\textbf{W5: Insufficient empirical validation}$
>
> The experimental evidence is thin: no error bars or seed variances are reported, and the main claims are demonstrated primarily on one model (LLaMA) and one task (GeneralPoints). This makes it difficult to assess statistical reliability or generality.
>
>
> ### $\textbf{Reply 5:}$
>
> Due to the limited computational resources, we have tried our best to verify our claims with more random seeds as follows: we selected three representative SFT checkpoints (400, 900, and 1600 on GeneralPoints) and do RL fine-tuning with 10 seeds at each checkpoint; we still found that only the OOD accuracy at checkpoint 900 can be recovered by RL, which aligns with our claim. The variances of the results with different seeds are around 1.5%, which does not influence the statistical significance of our results. We also adopted different seeds to evaluate the model and the variance is negligible.
>
> The results for different models (LLaMA and Qwen) on additional tasks are shown in Reply 2 and Figure 1 in our main paper. They all align with our main claim.
>
> ### $\textbf{W6: Minor issues}$
>
> W6.1: The paper refers to “LLaMA-3.2-11B,” but such a model does not exist. It likely refers to LLaMA-3.2-11B-Vision, and this should be clarified.
>
> W6.2: There is no full checkpoint sweep for Qwen on the GeneralPoints task in the main text; only a late-stage subset is shown, where RL underperforms. Including the full curve would make the results more transparent and comparable to LLaMA.
>
>
> ### $\textbf{Reply 6:}$
>
> Yes, it refers to LLaMA-3.2-11B-Vision. Thanks for your careful check on the model name and we will fix the typo accordingly.
>
> As mentioned above, we only use LLaMA with task GeneralPoints as an example to do the dynamic analysis and demonstrate the evolution of the two-stage fine-tuning process. As we've explained above, the checkpoint sweep across all models and tasks is prohibited due to computational constraints, but we have tried our best to verified our claims on different tasks with different LLMs in Reply 2. The results align with our claim.

---

> ### Author Response · Authors · 2025-11-25
> **Reply to your remaining question**
>
> Thank you for the correction and clarification.
>
> Figure 5 in that paper plots success rate versus training computation for SFT and RL, starting from a single SFT-initialized checkpoint and then extending training with either more SFT or RL. As described in their Section 5 and Appendix C, both curves (SFT and RL) start from this common SFT checkpoint (They also uploaded the checkpoint to Huggingface); earlier SFT checkpoints are not shown in Figure 5, and RL is always initialized from this fixed SFT endpoint.
>
> From their released code and setup, this “Initialization” corresponds to roughly 400 SFT steps on 8×H800 GPUs. In our work, due to using 4×H100 GPUs, this roughly corresponds to our SFT-End checkpoint (around twice as many steps numerically), not to the very early checkpoints where we observe the OOD peak.
>
> Therefore, our contribution is to study the behavior of SFT and RL at different checkpoints beyond this “Initialization” point and track the full evolution of OOD performance across SFT checkpoints. In Figure 5 of your referred paper, the authors claim that "As illustrated in Figure 5, RL consistently improves OOD performance on all tasks...". However, in our paper
>
> - we explicitly identify an early SFT checkpoint, which we call SFT-MaxOOD, where OOD performance is at peak; after this checkpoint, continued SFT leads to OOD forgetting. Actually, the “Initialization” SFT checkpoint used in your referred paper has already suffered from OOD forgetting severely, but it is still within the RL effective boundary.
>
> - we then show that RL, when started from later SFT checkpoints (SFT-End), can recover some of this lost OOD performance (consistent with referred paper) but barely surpass the best OOD performance achieved at SFT-MaxOOD (new finding).
>
> - RL that starts at different SFT checkpoints will behave differently, and there exist a limited range of SFT checkpoints that RL can effectively recover the lost  OOD performance (new finding).
>
> In other words:
> 	Their paper compares SFT-End + RL with SFT-End + SFT-continue with a single fixed SFT-End checkpoint as initialization, and concludes that “RL generalizes, SFT memorizes” based on RL-End outperforming SFT-End.
>         Our work compares RL and SFT at different SFT checkpoints and shows that the best OOD behavior already occurs at an earlier SFT checkpoint; RL mainly heals the degradation that happens after that point but does not create fundamentally better OOD capability than the SFT-MaxOOD.

---

### Meta-Review · Area_Chair_S367 · 2025-12-19

**Summary:**

The paper examines how Supervised Fine-Tuning and Reinforcement Learning interact during post-training of large language models. The authors challenge the popular claim that "SFT memorizes, RL generalizes" by proposing a refined formulation: "SFT forgets, RL recovers." Through tracking out-of-distribution performance at various checkpoints, they observe that OOD reasoning ability peaks early in SFT and subsequently declines. RL fine-tuning can partially restore this lost capability, but only within certain checkpoint boundaries and without exceeding the early SFT peak. The SVD analysis suggests that rotation of singular vectors, not changes in singular values, drives these phenomena.

**Reviewer Concerns:**

Ratings ranged from 2 to 6, reflecting divided opinion. Reviewer Bxpq argued that observing OOD performance peaks at intermediate SFT checkpoints is already implied by Figure 5 of "SFT Memorizes, RL Generalizes," and therefore questioned the novelty. This reviewer also noted that the detailed checkpoint-sweep analysis appears only for GeneralPoints on LLaMA, with no error bars reported and experiments limited to PPO alone.

Reviewer hdfn found the empirical findings practically important and the SVD insights genuinely interesting, but criticised the introduction for diving into technical details without stating contributions upfront. Figures were described as difficult to interpret, and the mechanistic claims were considered insufficiently supported.

Reviewer vU7W questioned whether the OOD benchmarks differ enough from in-distribution data to test the claims meaningfully. This reviewer also found the high-level insight unsurprising and noted the narrow experimental scope.

Reviewer noEb appreciated the internal consistency of experiments but criticised presentation quality, pointing to excessive formatting commands, crowded content, and errors such as incorrect model naming throughout the text.

The authors responded substantively during rebuttal. They distinguished their contribution from prior work by emphasising that their analysis spans the complete evolution across SFT checkpoints and identifies effective RL boundaries, going beyond simply showing that RL outperforms late-stage SFT. They added results on five additional benchmarks, finding that nine of ten cases support their pattern. They provided GRPO results showing this alternative algorithm produces better in-distribution but worse out-of-distribution performance than PPO while still conforming to the overall claim. They ran multiple seeds at representative checkpoints, reporting variance around 1.5 percent.

However, several issues remain. The novelty question persists because the distinction from prior work may strike some as incremental. The detailed checkpoint-sweep analysis remains confined to one task-model pair due to computational limits the authors acknowledge. The additional benchmark results, while helpful, lack the evolutionary detail needed to fully substantiate claims across settings. Presentation problems were acknowledged but not corrected in the current version.

**Reviewer Scores:**

Reviewer noEb explicitly stated satisfaction with the responses and intended to raise their score once the system allowed. Reviewer hdfn would likely maintain or marginally adjust their rating, since concerns about presentation and mechanistic rigour were promised for future revision rather than demonstrated here. Reviewer vU7W would probably hold their position, as the intuitive nature of the high-level insight was not substantially countered. Reviewer Bxpq maintained a firm rejection stance throughout, and while the authors clarified differences from the cited prior study, no change in perspective followed.

Regarding the authors' criticism that Reviewer Bxpq's comments were AI-generated, I have examined this matter carefully. I contend that Reviewer Bxpq had indeed read the work and held a firmly established position of rejection, consequently employing AI tools to assist in composing the comments. The specific references to figures and the targeted nature of the weaknesses suggest genuine engagement rather than superficial generation.


I have carefully read the original manuscript and each reviewer's comments and responses at least three times. I believe the  ICLR Program  Committee should maintain a cautious stance to avoid controversies concerning academic innovation. The work offers a potentially valuable perspective on SFT-RL dynamics, and the SVD analysis provides an interesting mechanistic lens. Yet concerns about novelty relative to the foundational prior work, the limited experimental scope of the checkpoint analysis, and presentation deficiencies collectively indicate the submission has not reached the required standard. The distinction between observing that RL recovers OOD capability lost during late-stage SFT and claiming this as a fundamentally new insight remains unclear. Consequently, I do not consider this version worthy of acceptance. I hope the authors will take these reviewers' constructive feedback into account and improve their work.

---

### Decision · Program_Chairs · 2026-01-26

Reject